# Why Masking Diffusion Works: Condition on the Jump Schedule for Improved Discrete Diffusion

**Alan N. Amin**
New York University
alanamin@nyu.edu

**Nate Gruver**
New York University
nvg7279@nyu.edu

**Andrew Gordon Wilson**
New York University
andrewgw@cims.nyu.edu

## Abstract

Discrete diffusion models, like continuous diffusion models, generate high-quality samples by gradually undoing noise applied to datapoints with a Markov process. Gradual generation in theory comes with many conceptual benefits; for example, inductive biases can be incorporated into the noising Markov process, and access to improved sampling algorithms. In practice, however, the consistently best performing discrete diffusion model is, surprisingly, masking diffusion, which does not denoise gradually. Here we explain the superior performance of masking diffusion by noting that it makes use of a fundamental difference between continuous and discrete Markov processes: discrete Markov processes evolve by discontinuous jumps at a fixed rate and, unlike other discrete diffusion models, masking diffusion *builds in the known distribution of jump times* and only learns where to jump to. We show that we can similarly bake in the known distribution of jump times into *any* discrete diffusion model. The resulting models — *schedule-conditioned diffusion* (SCUD) — generalize classical discrete diffusion and masking diffusion. By applying SCUD to models with noising processes that incorporate inductive biases on images, text, and protein data, we build models that outperform masking.

## 1 Introduction

Discrete diffusion models provide state-of-the-art conditional generation of discrete sequences. In biological sequence design, for example, they allow one to generate sequences flexibly conditioned on protein structure [15], DNA function [22], protein family [1], and other properties [9, 16]. They are also nearing state-of-the-art generation on language data [20].

A diffusion model is defined by a "forward" process, which gradually transforms data token-by-token into noise, and a "backward" transformation that turns noise into data, learned by optimizing an evidence lower bound (ELBO). In principle, the quality of the learned model should benefit from a forward process that captures structure in the data distribution. For example, works have suggested forward processes that are more likely to transform tokens into similar tokens – a more "gradual" noising process [2, 1] – as well as "state-dependent" processes that transform certain tokens more quickly than others [23]. Surprisingly, these methods are all outperformed by "masking diffusion" which has the simplest possible forward process – transforming each token into a masking token at a uniform rate [2, 1, 23]. Having seemingly arrived at the optimal forward process, work on discrete diffusion has instead shifted its focus to sampling [26, 18, 13, 7, 4] and scaling [15, 20].

Here we explore the causes of the superior performance of masking diffusion. We propose that masking diffusion benefits from a parameterization that forces the distribution of corruption or transition events, the "transition schedule", in the backward process to match the distribution in the forward process. Rather than allow us to conclude that masking is optimal, this insight allows us to expand the design space of discrete diffusion models to give any forward process the same property; we call models in this expanded design space *schedule conditioned diffusion* (SCUD). We show that

39th Conference on Neural Information Processing Systems (NeurIPS 2025).

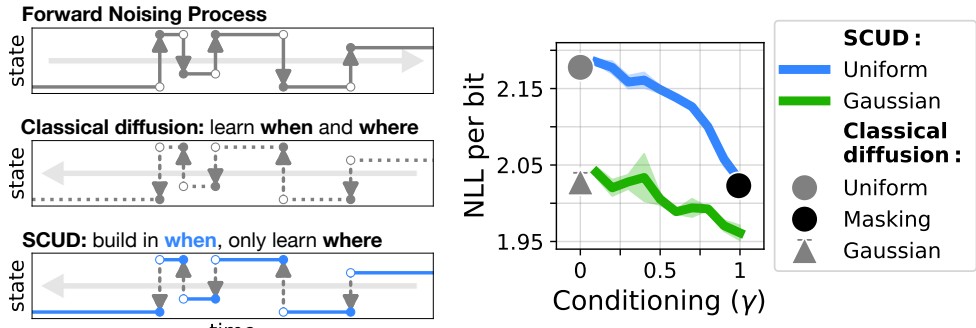

Figure 1: **Left: When trying to fit a forward process in diffusion, SCUD builds in the known distribution of transition times**. Classical discrete diffusion learns to reverse the paths generated in the forward process by learning both when and where to transition. SCUD instead builds in when to transition from the forward process, and therefore only must learn where to transition. **Right: Conditioning on more information about the transition time schedule results in a better likelihoods**. Models are fit on CIFAR-10 with $B = 128$ states. We show mean and standard error over 3 replicates. Details are in App. D.

applying SCUD to discrete diffusion with a uniform forward process, the result is masking diffusion, explaining its superior performance. We finally unlock the potential of structured and state-dependent discrete diffusion by building SCUD versions of these methods and see that they finally beat masking diffusion, achieving state-of-the-art results on images and proteins (Fig 1). We release our code at `https://github.com/AlanNawzadAmin/SCUD`.

## 2    Background

Our goal is to model data from a distribution $p(x_0)$ where $x_0$ is a sequence of discrete elements that belong to a set of size $B$. We consider the one-dimensional case first, and sequences later.

**Discrete diffusion**    In diffusion, we start with a distribution that is easy to sample from, $q(x_1)$; we then learn a parameterized Markov process from time 1 to time 0 that evolves samples from $q(x_1)$ to a distribution $q_\theta(x_0)$ that is approximately $p(x_0)$. To learn a Markov process that evolves $q(x_1)$ to $p(x_0)$, we first pick a simple Markov process that approximately evolves samples from $p(x_0)$ to $q(x_1)$ from time 0 to 1; then we try to match the trajectories from the parameterized Markov process $q_\theta((x_t)_{t\in[0,1]})$ that evolves "backward" from time 1 to 0 to those of the simple process $p((x_t)_{t\in[0,1]})$ that evolve "forward" from time 0 to 1 [3]. We do so by maximizing the evidence lower bound (ELBO)

$$
\begin{aligned}
E_{p(x_0)} \log q_\theta(x_0) \geq & E_{p((x_t)_{t\in[0,1]})} \log \frac{q_\theta((x_t)_{t\in[0,1]})}{p((x_t)_{t\in[0,1]}|x_0)} \\
= & E_{p((x_t)_{t\in[0,1]})} \log \frac{q_\theta((x_t)_{t\in[0,1]}|x_1)}{p((x_t)_{t\in[0,1]}|x_0,x_1)} + E_{p(x_1,x_0)} \log \frac{q(x_1)}{p(x_1|x_0)}.
\end{aligned}
\tag{1}
$$

This ELBO is maximized when the distribution of forward and backward trajectories match. The second term of the right hand side measures if the forward process indeed evolves samples $x_0 \sim p(x_0)$ to $q(x_1)$. The first term measures how well the forward and backward trajectories match.

**Discrete Markov processes and infinitesimal generators**    To define a diffusion model, we need to define a simple Markov process to generate $p((x_t)_{t\in[0,1]})$ and we need to parameterize the backward Markov process. Fortunately, discrete Markov processes are much easier to define than their continuous counterparts. Every time-homogeneous discrete Markov process is fully described by a $B \times B$ matrix that describes the "flow" of a particle at each instant in time known as the infinitesimal generator $\mathcal{L}$. In particular, $\mathcal{L}_{b,b'}$ describes the rate at which state $b$ transitions to state $b'$; the diagonal of $\mathcal{L}$ describes the rate of transitions out of $b$: $\mathcal{L}_{b,b} = -\sum_{b' \neq b} \mathcal{L}_{b,b'}$. Therefore, to simulate from a Markov process described by $\mathcal{L}$, starting at $x_t$, one simulates the time at which $x_t$ would transition to

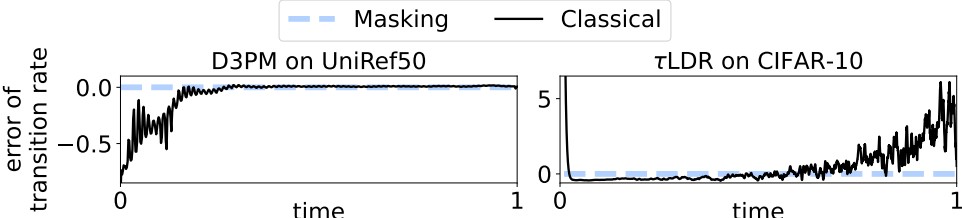

Figure 2: **State-of-the-art discrete diffusion models have backwards processes which do not match the forward process in when they transition.** Masking diffusion models have no such error as they build in the known forward rate into their backward process (dotted line). We plot the estimated transition rate of the backward process minus that of the forward process. We discuss details in App. D.

each other state $\Delta t_b \sim \text{Exp}(\mathcal{L}_{x_t,b})$ for $b \neq x_t$; then one transitions $x_t$ according to the first transition sampled: it take $\Delta t = \min_b \Delta t_b$ time to transition and $x_t$ transitions to $x_{t+\Delta t} = \text{argmin}_b \Delta t_b$. By a property of exponential distributions, the transition time is distributed according to the value on the diagonal of $\mathcal{L}$: $\Delta t \sim \text{Exp}(\sum_{b \neq x_0} \mathcal{L}_{x_0,b}) = \text{Exp}(-\mathcal{L}_{x_0,x_0})$. This procedure is known as the Gillespie algorithm [8].

**Picking the forward process** Two popular choices for the forward process are the uniform and masking processes. The uniform process has a constant rate of transitioning to any state. and the masking distribution has a constant rate of transition to a masking state $\emptyset$. Both of these processes are straightforward to simulate: simply sample $\Delta t \sim \text{Exp}(1)$, and then transition to a uniformly random state or to $\emptyset$. Other processes bake in inductive biases for text, images, and proteins [2, 1].

For typical Markov processes, information about the starting state $x_0$ becomes lost as $t$ gets larger and $p(x_t)$ gets closer to a stationary distribution $p(x_\infty)$. This distribution is a natural choice for $q(x_1)$ as long as $p(x_1|x_0)$ is close to converging to the stationary distribution.

In practice, $p(x_1|x_0)$ is usually not near $p(x_\infty)$, so we modulate the speed of the process by a rate $\beta_t$ at time $t$: at the instant $t$ we simulate from the process $\beta_t \mathcal{L}$. Simulating this modulated process for time $t$ is equivalent to simulating the original process for time $\int_0^t \beta_s ds$. By choosing $\beta_t$ to become large as $t \to 1$, we can be sure $p(x_1|x_0) \approx p(x_\infty) = q(x_1)$.

**Parameterizing the backward distribution** The backward Markov process is usually defined in terms of a parameterized, time-dependent, infinitesimal generator $\mathcal{L}_{\theta,t}$. The first term of Eqn. 1 is usually written as an integral in time $E_{t \sim \text{Unif}(0,1)} L(\mathcal{L}_{\theta,t}, t)$, for some $L$ which intuitively measures how well the $\mathcal{L}_{\theta,t}$ describes the "flow" of the reversal of $p((x_t)_t)$ at instant $t$ [3, 15].

## 3 Learning when and where to transition

To fit a discrete diffusion model, the backward process should match the forward in both *when* it transitions and *where* it transitions to. One should expect that learning where to transition is hard; on the other hand, since the distribution of when to transition is simple and known a priori in many cases, one should expect learning when to transition should be trivial. We see however in Fig. 2 that learning when to transition may also be challenging: state of the art published diffusion models have detectable differences in the transition rates of their forward and backward processes.

Unlike previously derived forms of the ELBO which are written as an integral of the discrepancy of the flow at each moment $t$, we will break up the ELBO into discrepancy of when and where to transition. Define the "transition schedule", $S = \{t_1, t_2, \ldots, t_M\}$, as the set of times at which $x_t$ transitions.

**Proposition 3.1.** *(Proof in Prop A.1 in the Appendix) The expression in Eqn. 1 is equal to the expression in Eqn 2 for some constant $C$.*

$$E_{p((x_t)_t)} \log \frac{q_\theta((x_t)_t|x_1, S)}{p((x_t)_t|x_0, x_1, S)} - \text{KL}(p(S)||q_\theta(S)) - E_{p(S,x_0)}\text{KL}(p(x_1|S, x_0)||q_\theta(x_1|S)) + C.$$

(2)

The first term represents the difference in log likelihoods between $q_\theta$ and $p$ when the transitions are known, which measures if the forward and backward processes match where they transition to. The second term measures if the forward and backward processes match when they transition. The third term, like the second term of Eqn. 1 intuitively measures if $p(x_1|x_0)$ has converged to $p(x_\infty)$.

To build diffusion models that better fit their objective, we therefore would like to incorporate knowledge of $p(S)$ into the model. Eqn 2 is suggestive of how to do this: set $q(S) = p(S)$ so that the second term becomes 0 and then learn where to transition by optimizing the first term. We call this procedure "schedule conditioning" (Fig. 1) and in Sec. 4 we describe how to perform it in practice.

Unlike diffusion models with the uniform forward process, diffusion models with the masking forward process are parameterized so that the distribution of times at which tokens are masked matches the distribution of times at which they are unmasked: these models know when to transition ( Fig. 2). In practice, masking diffusion models have been observed to outperform uniform diffusion models [2, 1, 14]. In Sec. 5 we will prove that applying our methods in Sec. 4 gives exactly masking diffusion, explaining their superior performance. By schedule conditioning other processes with more appropriate inductive biases, we also improve on masking diffusion (Fig 1).

## 4   Scheduled conditioned diffusion (SCUD)

In this section, motivated by Eqn. 2, we describe how to build discrete diffusion models that incorporate information about when to transition into a discrete diffusion model. We call such models Scheduled conditioned diffusion (SCUD) models.

Ideally we could set $q(S) = p(S)$; however, in general, $\mathcal{L}$ may not have constant transition rates at each state, in which case $S$ may be correlated with $x_0$ and $p(S)$ may be a complex distribution. Here we build a family of SCUD models that, instead of looking directly at transitions, introduce latent "events" which will act as transitions did above. These events occur with constant rate and often result in transitions; in some cases we discuss below, they will coincide exactly with transitions. We will condition on the schedule of these events, $S$.

In Sec. 4.1 we will describe models that condition on these event schedules, SCUD. Next in Sec. 4.2 we will write the loss in a form that is easy to train on high dimensional data. In App. B we will describe how we use standard techniques in discrete diffusion to parameterize our de-noising model, sample from SCUD, and choose the rate function $\beta_t$.

### 4.1   Conditioning on event schedules

**Markov processes with event schedules**   To sample from a uniform forward process starting at $x_t$, we sampled a transition time according to a rate that was independent of the current state, $\Delta t \sim \text{Exp}(1)$, and then sampled $x_{t+\Delta t}$ with uniform probability. Consider more generally the discrete Markov process on $x_t$ such that we sample an "event" $\Delta t \sim \text{Exp}(r)$, and then sample $x_{t+\Delta t} \sim \text{Categorical}(K_{x_t, \cdot})$ where $K_{x_t, \cdot}$ is a matrix whose rows are normalized distributions; note in this case $x_t$ may be equal to $x_{t+\Delta t}$. By appealing to the formal definition of $\mathcal{L}$, the next proposition tells us that this process has infinitesimal generator that flows according to the rate $r \times K$, with a $-I$ to describe the flow out of $x$.

**Proposition 4.1.** *(Proof in Prop A.2 in the Appendix) The infinitesimal generator of this process is $\mathcal{L} = r(K - I)$ where $I$ is the identity matrix. In particular, any Markov process with $\mathcal{L}$ can be simulated in the above way by picking an $r \geq \max_b -\mathcal{L}_{b,b}$ and setting $K = \mathcal{L}/r + I$.*

We note there are many choices of $r$ that allow one to write the same Markov process in this way. We will evaluate different choices in Sec. 5.

**Reversing the process conditioned on the event schedule**   Equip with the definition of events, we now try to show we can reverse the distribution of paths by just predicting what $x_t$ was before

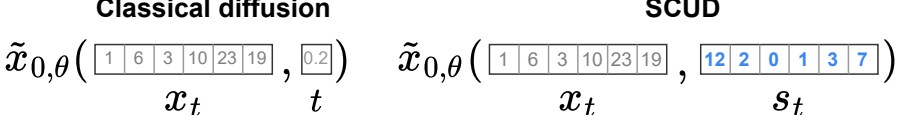

Figure 3: **When predicting the de-noised data $x_0$ from the noised-data $x_t$, SCUD uses more "fine-grained" information about the number of corruptions $s_t$ rather than just the corruption time $t$.** Classical discrete diffusion trains a model $\tilde{x}_{0,\theta}$ to predict the uncorrupted sequence $x_0$ from the corrupted sequence $x_t$ and information for how long it's been corrupted for $t$. SCUD trains a model to predict $\tilde{x}_{0,\theta}$ that replaces $t$ with more fine-grained information about how many corruptions have been applied to each token $s_t$, which be thought of as a measure of how "masked" each token is.

each event. Call $p((x_t)_t)$ the distribution of paths that start at $p(x_0)$ and evolve according to the above Markov process. The next proposition uses some algebra to suggest that we can simulate from $p((x_t)_t)$ "backwards" by 1) sampling the ending point $x_1 \sim p(x_1)$, 2) sampling the event schedule $\{t_1, t_2, \ldots, t_M\} \sim p(S)$, and then 3) going backwards, sampling where the particle came from at the $m$-th event.

**Proposition 4.2.** *(Proof in Prop A.3 in the Appendix) Call the event schedule $S = \{t_1, t_2, \ldots, t_M\}$ and $t_0 = 0$. Call $s_t$ the number of events up to time $t$, so $s_{t_m} = m$.*

$$p((x_t)_t, S) = p(S)p(x_1) \prod_{m=1}^{M} p(x_{t_{m-1}}|x_{t_m}, s_{t_m}). \tag{3}$$

We now aim to model this backwards process.

**SCUD: schedule conditioned discrete diffusion models**  As suggested in Sec 3, we wish to build a discrete diffusion model $q_\theta$ by setting $q(x_1) = p(x_\infty)$ and $q(S) = p(S)$. Prop. 4.2 suggests parameterizing $q$ so that, at each event, it predicts the previous state $x_{t_{m-1}}$ given 1) the current state $x_{t_m}$ and 2) the number of events that have occurred so far $s_t$. We call such a model a SCUD (schedule conditioned diffusion) model. With some algebra, in analogy with Eqn. 2, we get a closed form objective.

**Proposition 4.3.** *(Proof in Prop A.4 in the Appendix) Calling the event schedule $S = \{t_1, t_2, \ldots, t_M\}$ and $t_0 = 0$, $E_{p(x_0)} \log q_\theta(x_0)$ is greater than*

$$E \sum_{m=1}^{M} \mathrm{KL}(p(x_{t_{m-1}}|x_{t_m}, x_0, s_{t_m})||q_\theta(x_{t_{m-1}}|x_{t_m}, s_{t_m})) - E_{p(S,x_0)}\mathrm{KL}(p(x_1|s_1, x_0)||p(x_\infty)). \tag{4}$$

*where the first expectation is over $p((x_t)_t, S, x_0)$. This objective is minimized when $q_\theta(x_{t_{m-1}}|x_{t_m}, s_{t_m}) = p(x_{t_{m-1}}|x_{t_m}, s_{t_m})$.*

The first term teaches $q_\theta$ where to go at each event. The second term is small if $p(x_1)$ converges to $p(x_\infty)$. By Prop. 4.2 then, as the objective in Eqn. 4 is minimized, $q_\theta((x_t)_t)$ approaches $p((x_t)_t)$.

**Computing the objective**  The ELBO in Eqn. 4 is straightforward to compute. To calculate the first term, we note, writing each state as a one-hot vector,

$$p(x_{t_{m-1}}|x_{t_m}, x_0, s_t) = \frac{p(x_{t_{m-1}}|x_0, s_t)p(x_{t_m}|x_{t_{m-1}}, s_t)}{p(x_{t_m}|x_0, S)} = \frac{x_0^T K^{s_t-1} x_{t_{m-1}} x_{t_{m-1}}^T K x_{t_m}}{x_0^T K^{s_t} x_{t_m}}. \tag{5}$$

To calculate the second, we note $p(x_\infty)$ is the left eigenvector of $\mathcal{L}$ that corresponds to the eigenvalue 0 (as it does not change under flow from $\mathcal{L}$) and $p(x_1|s_1, x_0) = x_0^T K^{s_1} x_1$.

### 4.2   High dimensional data

For high dimensional data such as images, language, and biological sequences, it is common to choose processes $\mathcal{L}$ that act on each dimension independently. Say our data has $D$ dimensions $x_0^1, \ldots, x_0^D$ with each $x_0^d$ a discrete object in a set of size $B$. We extend SCUD to this case by simulating $D$ schedules for each dimension $S^1, \ldots, S^D \sim p(S)$; so $s_t$ becomes a $D$-dimensional vector (Fig. 3).

**Parameterizing** $q_\theta$  For a time $t$, if $s_t^d > 0$, define $\mathrm{pr}(x_t^d)$ as the state at the last event in dimension $d$, and $\mathrm{pr}(x_t)$ the previous state at each dimension; i.e., if the event schedule at dimension $d$ is $S^d = \{t_1^d, \ldots, t_m^d\}$ and $t \in [t_m^d, t_{m+1}^d)$, then $\mathrm{pr}(x_t^d) = x_{t_{m-1}^d}^d$. Our formula for reversing $p((x_t)_t)$ in Prop. 4.2 remains the same, but in App. B.2 we show $p(\mathrm{pr}(x_t)|x_t, s_t)$ factorizes. Thus we parameterize our predictor $q_\theta(\mathrm{pr}(x_t)|x_t, s_t)$ so it also factorizes as $\prod_{d=1}^D q_\theta(\mathrm{pr}(x_t^d)|x_t, s_t)$. Thus we get an objective as in Eqn. 4, but with a sum over $D$ in front.

**Efficient loss**  We could in principle use our objective in Eqn. 4 by taking empirical estimates of the expectation and the sum over events. In this case, however, each empirical sample corresponds to one event which affects a single dimension $d$, so it only checks the prediction $q_\theta(\mathrm{pr}(x_t^d)|x_t, s_t)$. The loss of other diffusion models, written as $E_{t \sim \mathrm{Unif}(0,1)} E_{x_t \sim p(x_t|x_0)} \sum_{d=1}^D L^d(\mathcal{L}_\theta, x_t, t|x_0, \mathcal{L})$, allow one to sample $t$ and then check the predictions of $q_\theta(\mathrm{pr}(x_t^d)|x_t, s_t)$ at that time for every $d$ in parallel. To write our objective in a similar form, we sample $t \sim \mathrm{Unif}(0, 1)$ and then add a weight $s_t^d \times \beta_t / \int_0^t \beta_s ds$ representing how likely an event is to occur at the instant $t$:

**Proposition 4.4.** *(SCUD loss) (Proof in Prop. A.6 in the Appendix) The first term of Eqn. 4 is*

$$- E_{t \sim \mathrm{Unif}(0,1)} E_{p(x_t, x_0, S)} \frac{\beta_t}{\int_0^t \beta_s ds} \times \sum_d s_t^d \mathrm{KL}(p(\mathrm{pr}(x_t^d)|x_t^d, s_t^d, x_0^d) || q_\theta(\mathrm{pr}(x_t^d)|x_t, s_t)). \quad (6)$$

We can approximate this objective by empirical estimates of all of the expectations and optimize with minibatch gradient descent. For a single evaluation of $q_\theta$ we can predict $\mathrm{pr}(x_t^d)$ for each dimension $d$ in parallel and check whether it matches the forward process along every dimension. The algorithm for calculating an estimate of the ELBO for a $x_0$ is summarized in App. B.1.

## 5 Connection SCUD to classical and masking discrete diffusion

To incorporate information about transitions into $q_\theta$, we wish to condition on the schedule. We described how conditioning on "events" in the previous section allow us to incorporate this structure. However not every event corresponds to a transition. The amount of information about the transitions that we bake into our model depends on the diagonal of $K$, the probabilities of no transition at an event. In turn the diagonal of $K$ will depend on our choice of the rate of events $r$. For a fixed $\mathcal{L}$, we can choose any rate $r \geq r^* = \max_b -\mathcal{L}_{b,b}$. Let's parameterize our choices of rate with a parameter $\gamma$: let $r = \gamma^{-1} r^*$. When $\gamma$ is 1, the rate of events is as slow as possible; when $\gamma \to 0$, the rate of events goes to $\infty$. $\gamma$ therefore represents the amount of schedule conditioning.

We now show that when $\mathcal{L}$ is uniform and $\gamma = 1 - 1/D$, that is, we nearly fully condition on the schedule, our process is equivalent to masking diffusion. On the other hand, as $\gamma \to 0$, we learn a backwards process while baking in no information about transitions; we show this recovers classical discrete diffusion.

### 5.1 Connection to masking diffusion

Say $\gamma = 1 - 1/D$ and $\mathcal{L}$ is uniform: $\mathcal{L}_{b,b'}$ is $1/B$ when $b \neq b'$. For this choice, $K$ is a matrix which has $1/B$ at every position. If a token is corrupted at least once by $K$ then it is distributed uniformly; it tell us nothing about $x_0$ so it is as if that token is "masked". When we condition on the event schedule, $s_t$ will tell us exactly which positions are masked when $s_t^d > 0$. By integrating out $s_t$ conditioned on the mask, we get exactly the masking diffusion objective [23].

**Proposition 5.1.** *(Proof in Prop. A.7 in the Appendix) Call the masking indicator $m_t^d = s_t^d > 0$. $\tilde{x}_{0,\theta}(x_t, s_t)$ only depends on $s_t$ through $m_t$. Defining $\alpha_t = \exp(-\int_0^t \beta_s ds)$, the objective Eqn. 6 is*

$$E_{t \sim \mathrm{Unif}(0,1) p(x_t, m_t|x_0)} \frac{\beta_t \alpha_t}{1 - \alpha_t} \sum_d x_0^T \log \tilde{x}_{0,\theta}(x_t, m_t)^d.$$

*The mask $m_t$ is distributed as $m_t^d \sim \mathrm{Bern}(1 - \alpha_t)$.*

In App. B.4 we also show that our choice for rate $\beta_t$ for this SCUD process is linear (in the sense $\alpha_t = 1 - t$), just as for the masking process as discussed in [2].

## 5.2 Connection to classical discrete diffusion

As $\gamma \to 0$, each event represents an infinitesimal change in $x_t$. Additionally, the number of events up to time $t$, $s_t$, grows larger but fluctuates less and less; inputting $s_t$ into $q_\theta(\mathrm{pr}(x_t^d)|x_t, s_t)$ becomes approximately identical to inputting the time $t$ into $q_\theta$. Therefore, as $\gamma \to 0$, $q_\theta$ predicts the infinitesimal change at time $t$: the infinitesimal generator. This is exactly the objective of classical discrete diffusion. The next proposition shows that when we take the limit $\gamma \to 0$ we recover exactly the loss from SEDD [15] which is also equivalent to that from $\tau$LDR [3].

**Proposition 5.2.** *(Proof in Prop. A.8 in the Appendix) As $\gamma \to 0$ the objective in Eqn. 6 converges to the SEDD loss (Eqn. 17).*

In App. B.4 we show that our choice for $\beta_t$ approaches that for classical discrete diffusion as $\gamma \to 0$.

# 6 Results

We apply SCUD to build an expanded design space for discrete diffusion models. We first demonstrate the results of Sec. 5 that SCUD with a uniform forward process interpolates between uniform and masking discrete diffusion. We next show that exploring the SCUD design space can unlock benefits from structured forward processes – we demonstrate state of the art model fits to images and protein data. Finally, with a case study in language, we discuss the practicality of SCUD – SCUD incurs minimal computational overhead, and, with some clever model decisions, can actually enable model decisions that were previously computationally intreactable. Other experimental details are in App. D.

## 6.1 Connection to other models

We show that by incorporating information about the distribution of transitions into a discrete diffusion model, one gets better fits to the forward process. We fit models to CIFAR-10 where each pixel takes a value from 1 to $B = 128$. In Fig. 1 we see that on this dataset discrete diffusion with a uniform forward process is outperformed by masking diffusion. We see that sweeping $\gamma$ between 0.1 and 1, SCUD with the uniform forward process interpolates the performance of the two models.

Next we build a structured forward process that builds in the inductive bias that similar pixel values describe similar colors – we set $\mathcal{L}_{i,j} = \exp(-200 \left(\frac{i-j}{B}\right)^2)$ for $i \neq j$, similar to the discrete-time Gaussian forward process in Austin et al. [2]. We see that a discrete diffusion model with this forward process slightly outperforms masking distribution. We next build SCUD models with this forward process; we see that these models better fit their objective as we incorporate more information about transitions – $\gamma \to 1$. These models outperform models that have structured forward processes (Gaussian) or those that just condition on the transition schedule (masking) without doing the other.

## 6.2 SCUD allows models to leverage structure in the forward process

Many previous discrete diffusion papers have noticed that masking is optimal or near-optimal across a wide range of domains, even outperforming bespoke forward process [1, 2]. Do structured forward process, which bake in inductive biases of the data, not help modeling? In this section we show that when we account for schedule conditioning ($\gamma = 1$), these bespoke processes actually substantially improve model fit on image and protein data. In particular, we achieve state-of-the-art diffusion model fits on these data by combining the schedule conditioning of masking with the strucuter of bespoke forward processes.

The structured forward processes we build for each modality will be inspired by those from Austin et al. [2] and Alamdari et al. [1]. However these papers used processes in discretized time that are not equivalent to and continuous time Markov process; thus we describe new structured processes for continuous time in terms of $\mathcal{L}$ or $K$.

In all cases we aim to make minimal modifications to the architecture and training from previous models so that differences in scores are due to schedule conditioning. We propose several techniques that enable moving from classical discrete diffusion to SCUD without substantial computational overhead, summarized in App. D.7. We also compare SCUD with our own re-implementations of classical and masking diffusion to best measure the effect of schedule conditioning and structure.

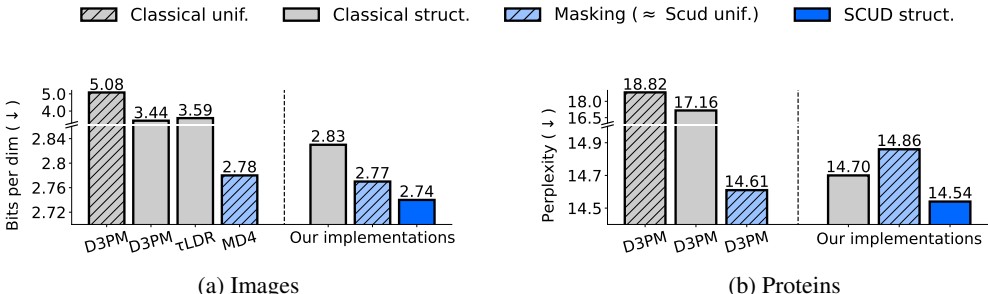

(a) Images  (b) Proteins

Figure 4: **Schedule conditioning unlocks improvements from structured forward processes on on images and proteins.** We compare discrete diffusion models from the literature and our own implementations and report model fit in bits per dimension on CIFAR-10 and Perplexity on Uniref50. Models labeled "unif." use a uniform forward process and those labelled "struct." use a structured forward process – a Gaussian prior for images and a BLOSUM prior for proteins (see App. C).

**Images** Here we build models on CIFAR-10 with $B = 256$ and compare to state of the art diffusion models. We use the architecture from [11] as in discrete diffusion models MD4 [23] and similar to that in D3PM [2] and $\tau$LDR [3]. To incorporate $s_t$ into our function, we replace additive layers that inject $t$ into every layer with FiLM [19] layers that incorporate $s_t$ into every layer. We also use the logistic parameterization from Salimans et al. [21] also used in D3PM, which interprets the output of the model as the parameters of a discretized logistic distribution over pixel values, so that similar pixel intensities have similar probabilities.

In Fig. 4a we compare SCUD with discrete diffusion models D3PM [2], $\tau$LDR [3], and MD4 [23] as well as our implementations of classical discrete diffusion models. We see that applying SCUD to model the Gaussian forward processes substantially improves likelihood with a fraction of the compute. Among previous discrete diffusion models, masking diffusion is the most performant despite not incorporating inductive biases. When accounting for schedule conditioning however, a structured process beats a uniform process (Scud str. beats masking / SCUD uniform); in particular, we achieve state-of-the-art ELBOs on images by unlocking the benefits of structured forward processes. This suggests that masking beats Gaussian diffusion in classical models because the benefit of schedule conditioning outweighs the benefit of incorporating inductive biases. By both incorporating inductive biases and schedule condition, SCUD unlocks the potential of Gaussian discrete diffusion on images.

In App. D.1 we examine samples from SCUD Gaussian. The samples resemble real objects much more than those from autoregressive models PixelCNN++ [21] and PixelSNAIL [5] which have state-of-the-art likelihoods. Furthermore, they contain clear objects like those from D3PM [2] or $\tau$LDR [3]; MD4 did not show or evaluate images. Future work could apply the many methods for improving sampling quality of discrete diffusion models to SCUD – we focus our exploration here on model fit as measured by likelihood.

**Proteins** Here we train models on the UniRef50 protein dataset with architectures from [1]. As in [1] we build a forward process using the BLOSUM matrix; this matrix describes the rates of mutations between amino acids seen in nature. We describe the details of our process in App C; we note $B = 31 = 20$ canonical amino acids $+ 11$ special tokens.

In Tab. 4b we compare SCUD BLOSUM with the small D3PM models from [1] as well as our implementations of classical discrete diffusion models. We see again that applying SCUD to uniform and BLOSUM diffusion substantially improves the model fit given a fraction of the compute budget (See App. D.6). In classical discrete diffusion, masking strongly outperforms BLOSUM diffusion; interestingly, in our reimplementation, classical BLOSUM outperformed masking. Nevertheless, by both schedule conditioning and incorporating inductive biases, SCUD BLOSUM outperforms masking and classical BLOSUM, achieving state-of-the-art model fit. In App. D.2 we show a similar result when starting from pre-trained weights.

| Method | Fwd. process | Perplexity |
|--------|--------------|------------|
| D3PM   | Masking      | 76.9*      |
| D3PM   | Graph        | 149.5*     |
| SCUD   | Masking      | 37.82      |
| SCUD   | Graph        | **37.63**  |

Table 1: **Schedule conditioning improves uniform corruptions and accommodates structured forward processes.** We compare to other discrete diffusion models on LM1B. *Numbers are not directly comparable with D3PM as a different tokenizer was used.

### 6.3 Scaling case study: Language

In all cases above, SCUD and classical diffusion had very similar run times – within 10% of each-other. This is because their loss computations are very similar and we only slightly changed the architecture of our neural network to accommodate SCUD; a full explanation is in App. D.7. Here we show that the SCUD design space also allows one to build models that are computationally intractable using classical discrete diffusion.

Above we looked at whether bespoke structured processes from previous works could help modeling. Austin et al. [2] build a discrete-time "nearest-neighbour graph" process on language and saw that it was substantially outperformed by masking; could we test if this process also helps modeling?

We must build models on the one billion words dataset with a $B = 30522$ vocabulary size. Unfortunately, the continuous-time classical diffusion loss requires calculating $\exp(t\mathcal{L})$, which is prohibitively expensive when $B = 30522$; indeed Luo et al. [15], Sahoo et al. [20] restrict $\mathcal{L}$ to be a uniform or masking process so they can calculate this quantity in closed form.

SCUD on the other hand only needs to compute $K^{s_t^d}$. We therefore define a sparse 10-nearest neighbour graph over the most frequent 2000 states, which make up 95% of tokens in the data. Our forward process diffuses along this graph with some probability or transitions approximately uniformly with some small probability; the less frequently used 25 thousand states always transition uniformly. We discuss the details in App C. The result is that $K$ is sparse so that $K^{s_t^d}$ can be efficiently calculated (computational complexity details in App. D.7).

Now able to implement the nearest neighbour process from Austin et al. [2] in continuous-time models, in Tab. 1 we compare the effect of incorporating this structure with SCUD with the results in Austin et al. [2]. In [2], masking strongly beats discrete diffusion with a nearest neighbor structure on this dataset (D3PM in Tab. 1). Unlike Austin et al. [2], accounting for schedule conditioning, when we add structure to the forward process, we improve our fit by a small amount. Unlike in images and proteins, the improvement is not substantial however, suggesting that this process could be improved.

## 7 Discussion

The choice of forward process is critical to the definition of a discrete diffusion model. Yet previous results have shown very strong performance from the simplest forward process — the masking process. SCUD offers an explanation for this: masking incorporates information about the transition schedule. By incorporating this information into models with other forward processes, SCUD allowed us to build models that build in inductive biases and outperform masking.

**Condition on everything?**  By adding more information, $S$, to the backwards process, SCUD was able to fit the process much better. Future work may explore adding or removing information from $S$: for forward process with multiple types of mutations, $S$ may count each type of mutation; on the other hand, SCUD trains a de-noiser $q_\theta$ that conditions on $S$ – different applications may call to learn de-noiser that takes in different information $S$.

Is there a limit to how much information we can put in $S$? As more information goes into $S$, the inference task for $\tilde{x}_{0,\theta}$ becomes more challenging. As well, the third term in Eqn. 2 may become large – if $S$ describes exactly every mutation in the forward process for example, then $p(x_1|S, x_0)$ is a point mass, it never converges. Indeed, one may consider extending SCUD to continuous diffusion by setting $S$ to be, by analogy, the distance $x_t$ has traveled from $x_0$; but in this case, $p(x_1|S, x_0)$

has support only on a sphere – it doesn't converge. In our case, the $S$ we considered was able to overcome these challenges to improve model fit, but future work may need to consider this tradeoff when navigating the modeling space.

**Sampling from diffusion models**  This work focuses on achieving better ELBOs without investigating the effect on sampling. There is huge work on exploring sampling from discrete diffusion models. These methods approach the problem of trying to get a more faithful sample of $q_\theta(x_t)$ [3, 26] or combining the de-noiser $q_\theta$ with other predictors to get better samples [18, 13]; another popular family of methods that falls in this camp is discrete flow matching [7, 4] – we explain how to extend flow matching to SCUD in App. E. There has however been comparatively little work exploring the design space of the forward process, potentially because of a mistaken belief that masking could be the optimal forward process [23] (a telling exception is learning masking forward processes in Shi et al. [23] and Wang et al. [25]). SCUD shows the potential of improving model fit by exploring improved forward processes, potentially complementing work on sampling in future work.

## Acknowledgements

We thank Lily Li and Marc Finzi for helpful feedback. This work was supported in part by NSF CAREER IIS-2145492, NSF CDS&E- MSS 2134216, NSF HDR-2118310, BigHat Biosciences, and Capital One.

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

# A  Proofs of results

**Proposition A.1.** *(Proof of Prop 3.1) The expression in Eqn. 1 is equal to the expression in Eqn 2 for some constant $C$.*

*Proof.* $S$ is a deterministic function of $(x_t)_t$ so we can write the first term of Eqn. 1 as

$$E_{p((x_t)_{t\in[0,1]}|x_0)} \log \frac{q_\theta((x_t)_{t\in[0,1]}|x_1)}{p((x_t)_{t\in[0,1]}|x_0,x_1)} = E_{p((x_t)_{t\in[0,1]}|x_0)} \log \frac{q_\theta((x_t)_{t\in[0,1]},S|x_1)}{p((x_t)_{t\in[0,1]},S|x_0,x_1)}$$

$$= E_{p((x_t)_{t\in[0,1]}|x_0)} \log \frac{q_\theta((x_t)_{t\in[0,1]}|x_1,S)}{p((x_t)_{t\in[0,1]}|x_0,x_1,S)}. \quad (7)$$

$$+ E_{p(S,x_1|x_0)} \log \frac{q_\theta(S|x_1)}{p(S|x_0,x_1)}.$$

We can combine the second term of this equation with the second term of Eqn. 1 to get

$$E_{p(S,x_1|x_0)} \log \frac{q_\theta(S|x_1)}{p(S|x_0,x_1)} + E_{p(x_1|x_0)} \log \frac{q(x_1)}{p(x_1|x_0)}$$

$$= E_{p(S|x_0)} \log \frac{q_\theta(S)}{p(S|x_0)} + E_{p(S,x_1|x_0)} \log \frac{q_\theta(x_1|S)}{p(x_1|x_0,S)} \quad (8)$$

$$= E_{p(S|x_0)} \log \frac{q_\theta(S)}{p(S)} + E_{p(S|x_0)} \log \frac{p(S)}{p(S|x_0)} - E_{p(S|x_0)} \mathrm{KL}(p(x_1|x_0,S)|q_\theta(x_1|S)).$$

The first term is $-\mathrm{KL}(p(S)||q_\theta(S))$ and the second does not depend on $q$. This completes the proof. $\square$

**Proposition A.2.** *(Proof of Prop 4.1) The infinitesimal generator of this process is $\mathcal{L} = r(K - I)$ where $I$ is the identity matrix. In particular, any Markov process with $\mathcal{L}$ can be simulated in the above way by picking an $r \geq \max_b -\mathcal{L}_{b,b}$ and setting $K = \mathcal{L}/r + I$.*

*Proof.* The process is described is clearly Markov. By the formal definition of $\mathcal{L}$, for $b' \neq b$,

$$\mathcal{L}_{b,b'} = \lim_{t\to 0} \frac{1}{t} p(x_t = b'|x_0 = b)$$

$$= \lim_{t\to 0} \frac{1}{t} (p(\text{an event occurs before } t) \times p(\text{the event transitions to } b') + o(t)) \quad (9)$$

$$= \lim_{t\to 0} \frac{1}{t}(1 - e^{-rt})K_{b,b'} = rK_{b,b'}.$$

Then, since the rows of $K$ sum to 1,

$$\mathcal{L}_{b,b} = -\sum_{b'\neq b} \mathcal{L}_{b,b'} = -r\sum_{b'\neq b} K_{b,b'} = -r(1 - K_{b,b}).$$

The second statement follows from rearranging the first. The requirement that $r \geq \max_b -\mathcal{L}_{b,b}$ comes from the fact that all entries in $K$ must be non-negative and $K_{b,b} = \mathcal{L}_{b,b}/r + 1$. $\square$

**Proposition A.3.** *(Proof of Prop 4.2 in the Appendix) Call the event schedule $S = \{t_1, t_2, \ldots, t_M\}$ and $t_0 = 0$. Call $s_t$ the number of events up to time $t$, so $s_{t_m} = m$.*

$$p((x_t)_t, S) = p(S)p(x_1) \prod_{m=1}^{M} p(x_{t_{m-1}}|x_{t_m}, s_{t_m}). \quad (10)$$

*Proof.*

$$p((x_t)_t, S) = p(S)p(x_1)p(x_{t_{0:M}}|x_1, S)$$

$$= p(S)p(x_1) \prod_{m=1}^{M} p(x_{t_{m-1}}|x_{t_{m:M}}, S).$$

By the Markov property, $p(x_{t_{m-1}}|x_{t_{m:M}}, S) = p(x_{t_{m-1}}|x_{t_m}, S)$. Finally, $p(x_{t_{m-1}}|x_{t_m}, S) \propto p(x_{t_m}|x_{t_{m-1}}, S)p(x_{t_{m-1}}|S) = p(x_{t_m}|x_{t_{m-1}})p(x_{t_{m-1}}|s_{t_{m-1}})$ only depends on $S$ through $s_{t_{m-1}}$, or equivalently, $s_{t_m} = 1 + s_{t_{m-1}}$. $\square$

**Proposition A.4.** *(Proof of Prop. 4.3) Calling the event schedule $S = \{t_1, t_2, \ldots, t_M\}$ and $t_0 = 0$.*

$$E_{p(x_0)} \log q_\theta(x_0) \geq - E_{p((x_t)_t, S, x_0)} \sum_{m=1}^{M} \text{KL}(p(x_{t_{m-1}}|x_{t_m}, x_0, s_{t_m}) || q_\theta(x_{t_{m-1}}|x_{t_m}, s_{t_m})) \tag{11}$$
$$- E_{p(S, x_0)} \text{KL}(p(x_1|s_1, x_0) || p(x_\infty)).$$

*This objective is minimized when $q_\theta(x_{t_{m-1}}|x_{t_m}, s_{t_m}) = p(x_{t_{m-1}}|x_{t_m}, s_{t_m})$.*

*Proof.* Just as with the classical ELBO, we can write

$$E_{p(x_0)} \log q(x_0) \geq E_{p(x_0, S)} E_{p((x_t)_{t \in [0,1]}, S|x_0)} \log \frac{q_\theta((x_t)_{t \in [0,1]}, S)}{p((x_t)_{t \in [0,1]}, S|x_0)}. \tag{12}$$

Then we can break it up as in Prop. A.1 to get

$$E_{p(x_0)} \log q(x_0) \geq E_{p((x_t)_t)} \log \frac{q_\theta((x_t)_t|x_1, S)}{p((x_t)_t|x_0, x_1, S)} - \text{KL}(p(S)||q(S))$$
$$E_{p(S|x_0)} \log \frac{p(S)}{p(S|x_0)} - E_{p(S, x_0)} \text{KL}(p(x_1|S, x_0) || q_\theta(x_1|S)). \tag{13}$$

By our definition of the event schedule and $q(S)$, the second and third term on the right are 0. For the fourth term, clearly $p(x_1|x_0, S) = p(x_1|x_0, s_1)$.

By our definition of $q_\theta$,

$$q_\theta((x_t)_t|x_1, S) = \prod_{m=1}^{M} q(x_{t_{m-1}}|x_{t_m}, s_{t_m}).$$

As in the proof of Prop. A.3, we can write

$$p((x_t)_t|x_0, x_1, S) = \prod_{m=1}^{M} p(x_{t_{m-1}}|x_0, x_1, S, x_{t_{m:M}}) = \prod_{m=1}^{M} p(x_{t_{m-1}}|x_0, s_{t_m}, x_{t_m})$$

where the last equality follows by the Markov property. Thus the first term is

$$\sum_{m=1}^{M} \log \frac{q(x_{t_{m-1}}|x_{t_m}, s_{t_m})}{p(x_{t_{m-1}}|x_0, s_{t_m}, x_{t_m})} = - \sum_{m=1}^{M} \text{KL}(p(x_{t_{m-1}}|x_0, s_{t_m}, x_{t_m}) || q(x_{t_{m-1}}|x_{t_m}, s_{t_m})).$$

$\square$

**Proposition A.5.** *(Proof of Prop B.1) $p(x_t|x_t, x_0, s_t)$ factorizes as $\prod_{d=1}^{D} p(\text{pr}(x_t^d)|x_t^d, x_0^d, s_t^d)$ and, when marginalizing over $x_0$, each dimension of $x_{t_{m-1}}$ is independent:*

$$p(\text{pr}(x_t)|x_t, s_t) = \prod_{d=1}^{D} p(\text{pr}(x_t^d)|x_t, s_t).$$

*Proof.*

$$p(\text{pr}(x_t)|x_t, x_0, s_t) = \frac{p(\text{pr}(x_t^d)|x_0, s_t)p(x_t|\text{pr}(x_t^d))}{p(x_t|x_0, s_t)} = \prod_{d=1}^{D} \frac{p(\text{pr}(x_t^d)|x_0^d, s_t^d)p(x_t^d|\text{pr}(x_t^d))}{p(x_t^d|x_0^d, s_t)}$$

which equals $\prod_{d=1}^{D} p(\text{pr}(x_t^d)|x_t^d, x_0^d, s_t^d)$. The second claim follows from integrating the later expression. $\square$

**Proposition A.6.** *(Proof of Prop. 4.4) Define, if $s_t^d > 0$, $\text{pr}(x_t^d)$ as the state at the last event in dimension $d$. Then the first term of Eqn. 4 is*

$$- E_{t \sim \text{Unif}(0,1)} E_{p(x_t, x_0, S)} \frac{\beta_t}{\int_0^t \beta_s ds} \sum_d s_t^d \text{KL}(p(\text{pr}(x_t^d)|x_t^d, s_t^d, x_0^d) || q_\theta(\text{pr}(x_t^d)|x_t, s_t)). \tag{14}$$

*Proof.* Call $S^d = \{t_1^d, \ldots, t_{M^d}^d\}$. The first term of Eqn. 4 can be written as

$$-E_{p((x_t)_t, S, x_0)} \sum_{d=1}^{D} \sum_{m=1}^{M^d} \mathrm{KL}(\mathrm{pr}(x_{t_m^d}^d)|x_{t_m^d}^d, x_0^d, s_{t_m^d}^d)||q_\theta(\mathrm{pr}(x_{t_m^d}^d)|x_{t_m^d}, s_{t_m^d})).$$

The term in the sum can be written as $L(s_t, x_t, x_0, d)$ so we can write

$$E_{p((x_t)_t, S, x_0)} \sum_{d=1}^{D} \sum_{t \in S^d} L(s_t, x_t, x_0, d) = \sum_{d=1}^{D} E_{p(S^d)} \sum_{t \in S^d} E_{p(x_0)p(S^{-d})p(x_t|x_0, s_t)} L(s_t, x_t, x_0, d).$$

Call the function after $\sum_{t \in S^d}$ equal to $C(t, s_t^d)$ so we can write the loss as $E_{p(S^d)} \sum_{t \in S^d} C(t, s_t^d)$. We now investigate the measure $E_{p(S^d)} \sum_{t \in S^d}$. First note that $E_{p(S^d)} \sum_{t \in S^d}$ is clearly absolutely continuous in $t$ with respect to the Lebesgue measure so this expression can be written as $E_{t \sim \mathrm{Unif}(0,1)} \sum_{s_t^d} f(t, s_t^d) C(t, s_t^d)$ for some function $f$. By the Lebesgue differentiation theorem, almost everywhere,

$$f(t', s) = \lim_{\epsilon \to 0} E_{p(S^d)} \sum_{t \in S^d} \mathbb{1}(t \in [t' - \epsilon, t'], s_{t'}^d = s)/\epsilon$$

$$= p(s_{t'}^d = s) \lim_{\epsilon \to 0} E\left[ \# \text{ events in } [t' - \epsilon, t'] | s_{t'}^d = s \right] / \epsilon. \tag{15}$$

The distribution of events on an interval $[0, t]$ is a Poisson process with density $\mu(s) = r\beta_s$; we can simulate this by drawing $s_t \sim \mathrm{Pois}(\int_0^t \beta_s ds)$ and then distributing the $s_t^d$ events with probability according to $\mu/\mu([0, t])$. Therefore, conditioned on $s$ events occurring on $[0, t']$, the density of events occurring at $[t' - \epsilon, t']$ is $\mu(t')/\mu([0, t'])$, that is, the expectation in Eqn. 15 is

$$s \text{ events} \times \frac{\mu(t')}{\mu([0, t'])} \text{ mass} = s \frac{\beta_{t'}}{\int_0^{t'} \beta_s ds}.$$

Subbing this into the previous equation completes the proof. $\qquad\square$

**Proposition A.7.** *(Proof of Prop. 5.1) Defining $\alpha_t = \exp(-\int_0^t \beta_s ds)$, the objective in Eqn. 6 is*

$$E_{t \sim \mathrm{Unif}(0,1)} E_{p(m_t)} E_{p(x_t|x_0, m_t)} \frac{\beta_t \alpha_t}{1 - \alpha_t} \sum_d x_0^T \log \tilde{x}_{0,\theta}(x_t, m_t)^d.$$

*Proof.* If $s_t^d > 1$ then $\mathrm{pr}(x_t^d)$ is corrupted so $p(\mathrm{pr}(x_t^d)|x_t^d, s_t^d, x_0^d)$ is a uniform categorical and doesn't depend on $x_0$; therefore, by our parameterization of $q_\theta$, we have that the KL term in the loss Eqn 6 is non-zero if and only if $s_t^d = 1$. As well, when $s_t^d = 1$, $p(\mathrm{pr}(x_t^d)|x_t^d, s_t^d = 1, x_0^d) = \delta_{x_0}$. In this case we can write the loss as

$$E_{t \sim \mathrm{Unif}(0,1)} \frac{\beta_t}{\int_{s < t} \beta_s ds} E_{p(S)} E_{p(x_t|S, x_0)} \sum_d \mathbb{1}(s_t^d = 1) x_0^T \log \tilde{x}_{0,\theta}(x_t, s_t)^d.$$

Finally note that when $\tilde{x}_{0,\theta}(x_t, s_t)$ predicts $x_0$, $s_t$ is only useful in telling the model which tokens are corrupted. If we call $m_t = s_t > 0$ an indicator of which tokens have been corrupted, then we can parameterize our prediction as $\tilde{x}_{0,\theta}(x_t, m_t)$.

Note $p(x_t|x_0, S) = p(x_t|x_0, m_t)$, so

$$E_{p(S)} E_{p(x_t|S, x_0)} \sum_d \mathbb{1}(s_t^d = 1) x_0^T \log \tilde{x}_{0,\theta}(x_t, m_t)^d =$$

$$E_{p(m)} E_{p(x_t|m_t, x_0)} \sum_d p(s_t^d = 1|m_t^d) x_0^T \log \tilde{x}_{0,\theta}(x_t, m_t)^d. \tag{16}$$

$s_t \sim \mathrm{Pois}(\int_0^t \beta_s ds)$ so $p(s_t^d = 1|m_t^d) = 0$ if $m_t^d = 0$ and

$$p(s_t^d = 1|m_t^d) = p(s_t^d = 1|s_t^d \geq 1) = \frac{\int_0^t \beta_s ds \, \alpha_t}{1 - \alpha_t}.$$

$\qquad\square$

**Proposition A.8.** *(Proof of Prop. 5.2) Define the score function estimator as in SEDD [15][1]*

$$\tilde{s}(x_t,t)^d_{\theta,b} = \frac{q_\theta(x_t^d = b|x_t^{-d})}{q_\theta(x_t^d|x_t^{-d})} = \frac{E_{\tilde{x}_{0,\theta}(x_t,s_t)}p(x_t^d = b|x_0^d)}{E_{\tilde{x}_{0,\theta}(x_t,s_t)}p(x_t^d|x_0^d)}.$$

*Suppressing the dependence of $\tilde{s}_\theta$ on $x_t, t$, as $\gamma \to 0$ the objective in Eqn. 6 converges to*

$$-E_{t\sim\text{Unif}(0,1)}E_{p(x_0,x_t)}\beta_t \sum_d \left[ \sum_{b\neq x_t^d} \mathcal{L}_{b,x_t^d}\left( \tilde{s}^d_{\theta,b} - \frac{p(x_t^d = b|x_0^d)}{p(x_t^d|x_0^d)}\log\tilde{s}^d_{\theta,b} - g\left(\frac{p(x_t^d = b|x_0^d)}{p(x_t^d|x_0^d)}\right) \right) \right] \tag{17}$$

*where $g(x) = x(\log x - 1)$.*

*Proof.* Note $s_t \sim \text{Pois}(r^* \int_0^t \beta_s ds/\gamma)$, so, as $\gamma \to 0$, $s_t\gamma$ converges to $r^* \int_0^t \beta_s ds$.
As $\gamma \to 0$,

$$K^{s_t} = (I + \gamma\mathcal{L}/r^*)^{s_t} = \exp(\gamma s_t\mathcal{L}/r^*) + o(\gamma) \to \exp\left(\int_0^t \beta_s ds\mathcal{L}\right) = Q_t,$$

where $Q_t$ is the matrix where $Q_{t,b,b'} = p(x_t = b'|x_0 = b)$.

$$\begin{aligned}
q_\theta(\text{pr}(x_t^d)|x_t,s_t) &= \frac{Kx_t^d \circ K^{s_t-1}\tilde{x}^d_{0,\theta}}{x_t^{d,T}K^{s_t}\tilde{x}^d_{0,\theta}} \\
&= \frac{Kx_t^d \circ K^{-1}Q_t\tilde{x}^d_{0,\theta}}{x_t^{d,T}Q_t\tilde{x}^d_{0,\theta}} + o(\gamma) \\
&= Kx_t^d \circ K^{-1}\tilde{s}^d_\theta + o(\gamma) \\
&= x_t^d + \frac{\gamma}{r^*}\left(\mathcal{L}x_t^d \circ \tilde{s}^d_\theta - x_t^d \circ \mathcal{L}\tilde{s}^d_\theta\right) + o(\gamma).
\end{aligned} \tag{18}$$

The expression for $p(\text{pr}(x_t^d)|x_t^d, x_0^d, s_t^d)$ is identical replacing $\tilde{x}_{0,\theta}$ with $x_0$. Thus

$$\begin{aligned}
&-\text{KL}(p(\text{pr}(x_t^d)|x_t^d, s_t^d, x_0^d)||q_\theta(\text{pr}(x_t^d)|x_t,s_t)) \\
&= \sum_{b\neq x_t^d} \frac{\gamma}{r^*}\mathcal{L}_{b,x_t^d}\frac{p(x_t^d = b|x_0^d)}{p(x_t^d|x_0^d)}\log\frac{\tilde{s}^d_{\theta,b}}{p(x_t^d = b|x_0^d)/p(x_t^d|x_0^d)} \\
&\quad + (1-O(\gamma))\log\frac{1 + \frac{\gamma}{r^*}\left(\mathcal{L}_{x_t^d,x_t^d} - x_t^d\mathcal{L}\tilde{s}^d_\theta\right)}{1 + \frac{\gamma}{r^*}\left(\mathcal{L}_{x_t^d,x_t^d} - x_t^d\mathcal{L}(p(x_t^d = b|x_0^d)/p(x_t^d|x_0^d))_b\right)} + o(\gamma) \\
&= \frac{\gamma}{r^*}\left[ \sum_{b\neq x_t^d} \mathcal{L}_{b,x_t^d}\left( \tilde{s}^d_{\theta,b} - \frac{p(x_t^d = b|x_0^d)}{p(x_t^d|x_0^d)}\log\tilde{s}^d_{\theta,b} - g\left(\frac{p(x_t^d = b|x_0^d)}{p(x_t^d|x_0^d)}\right) \right) \right] + o(\gamma).
\end{aligned} \tag{19}$$

Multiplying this by $s_t^d$, we get $\frac{\gamma}{r^*}s_t^d \to \int_0^t \beta_s ds$. □

# B Details of method

Here we describe how we parameterize, sample, and pick $\beta_t$ for SCUD.

## B.1 Algorithm for estimating ELBO

We calculate $p(x_\infty)$ from an spectral decomposition of $K$, or, if $K$ is very large, using power iteration.

---

[1]Recall $\tilde{x}^d_{0,\theta}(x_t, s_t)$ is trained to fit $p(x_0^d|x_t^{-d}, s_t^{-d})$.

---

**Algorithm 1** Unbiased estimate of the SCUD ELBO (Eqn. 4) using Prop. 4.4

---

**Input:** $x_0$
$S \sim p(S)$
$t \sim \mathrm{Unif}(0,1)$
// Sample $x_t$
**for** $d = 1, \ldots, D$ **do**
    $x_t^d \sim \mathrm{Categorical}(K^{s_t^d} x_0^d)$
**end for**
// Denoise one event of each dimension of $x_t$
Predict $\tilde{x}_{0,\theta}(x_t, s_t)$
**for** $d = 1, \ldots, D$ **do**
    Calculate $q_\theta(\mathrm{pr}(x_t^d)|x_t^d, s_t^d)$                                      ▷ use Eqn. 20
    Calculate $p(\mathrm{pr}(x_t^d)|x_t^d, s_t^d, x_0^d)$                                  ▷ use Eqn. 5
    Calculate $p(x_1^d|s_1^d, x_0^d) = \mathrm{Categorical}(K^{s_1^d} x_0^d)$.
**end for**
**Return:**
$$-\sum_{d=1}^{D} \left( \frac{s_t^d \beta_t}{\int_0^t \beta_s ds} \mathrm{KL}(p(\mathrm{pr}(x_t^d)|x_t^d, s_t^d, x_0^d) || q_\theta(\mathrm{pr}(x_t^d)|x_t^d, s_t^d)) + \mathrm{KL}(p(x_1^d|s_1^d, x_0^d) || p(x_\infty)) \right).$$

---

## B.2 Parameterization

$q_\theta$ must predict, for each dimension, $p(\mathrm{pr}(x_t^d)|x_t, s_t)$, which is an expectation over the posterior of $x_0^d$ given $x_t$ and $S$:

$$\sum_{x_0^d} p(\mathrm{pr}(x_t^d)|x_t^d, s_t^d, x_0^d) p(x_0^d|x_t, S) = \sum_{x_0^d} p(x_t^d|\mathrm{pr}(x_t^d)) p(\mathrm{pr}(x_t^d)|s_t^d, x_0^d) \frac{p(x_0^d|x_t, s_t)}{p(x_t^d|s_t^d, x_0^d)}.$$

Below we show that the fraction on the right hand side is proportional to $p(x_0^d|x_t^{-d}, s_t^{-d})$ where $x_t^{-d}$ and $s_t^{-d}$ are $x_t$ and $s_t$ without dimension $d$. Other discrete diffusion methods parameterize their $q_\theta$ to predict analogues of this quantity. Austin et al. [2] predicted a similar quantity rather than directly predicting $p(x_0^d|x_t, S)$. Moreover, predicting $p(x_0^d|x_t^{-d}, s_t^{-d})$ is identical to predicting $p(x_0^d|x_t, S)$ when $x_t^d$ is masked. Predicting this quantity has the benefit that we do not need to learn what $x_t^d$ tells us about $x_0^d$; it is baked into our prediction. We parameterize our $q_\theta$ similarly.

Thus, to predict $q_\theta(\mathrm{pr}(x_t^d)|x_t, S)$ we input $x_t$ and $s_t$ into a neural network that outputs a vector of probabilities $\tilde{x}_{0,\theta}$ and set $q_\theta(\mathrm{pr}(x_t^d)|x_t, s_t)$ equal to

$$\sum_b p(x_t^d|\mathrm{pr}(x_t^d)) p(\mathrm{pr}(x_t^d)|s_t^d, x_0^d = b) \tilde{x}_{0,\theta,b} \tag{20}$$

which can be writen as $K x_t^d \circ K^{s_t^d - 1, T} \tilde{x}_{0,\theta}$. Note we do not explicitly forbid $\tilde{x}_{0,\theta}$ from using $x_t^d, s_t^d$ to predict $x_0^d$.

Now we show that $p(\mathrm{pr}(x_t^d)|x_0, s_t, x_t)$ factorizes across its dimensions.

**Proposition B.1.** *(Proof in Prop A.5 in the Appendix)* $p(x_t|x_t, x_0, s_t)$ *factorizes as* $\prod_{d=1}^{D} p(\mathrm{pr}(x_t^d)|x_t^d, x_0^d, s_t^d)$ *and, when marginalizing over* $x_0$, *each dimension of* $x_{t_{m-1}}$ *is independent:*

$$p(\mathrm{pr}(x_t)|x_t, s_t) = \prod_{d=1}^{D} p(\mathrm{pr}(x_t^d)|x_t, s_t).$$

Recall this allows us to parameterize $q_\theta(\mathrm{pr}(x_t)|x_t, s_t)$ so it also factorizes as $\prod_{d=1}^{D} q_\theta(\mathrm{pr}(x_t^d)|x_t, s_t)$.

We parameterize $q_\theta(\mathrm{pr}(x_t^d)|x_t, s_t)$ to predict

$$\frac{p(x_0^d|x_t, s_t)}{p(x_t^d|x_0^d, s_t^d)} = \frac{p(x_t^d, s_t^d|x_0^d, x_t^{-d}, s_t^{-d})}{p(x_t^d|x_0^d, s_t^d) p(x_t^d, s_t^d)} p(x_0^d|x_t^{-d}, s_t^{-d}) = \frac{p(x_t^d|x_0^d, x_t^{-d}, s_t) p(s_t^d)}{p(x_t^d|x_0^d, s_t^d) p(x_t^d, s_t^d)} p(x_0^d|x_t^{-d}, s_t^{-d}).$$

Now note $p(x_t^d | x_0^d, x_t^{-d}, s_t) = p(x_t^d | x_0^d, s_t)$ and $p(s_t^d)/p(x_t^d | s_t^d)$ does not depend on $x_0^d$. Thus we aim to predict a quantity proportional to $p(x_0^d | x_t^{-d}, s_t^{-d})$; we call our prediction $\tilde{x}_{0,\theta}(x_t, s_t)$, which we plug into Eqn. 20 and then normalize.

## B.3 Sampling

To sample, in principle we could take $x_1 \sim p(x_\infty)$, $S \sim p(S)$, and then iteratively reverse each event in $S$ in order using our predictions of $q_\theta(\mathrm{pr}(x_t^d) | x_t, s_t)$. For data with many dimensions however, $S$ could contain tens of thousands of events, requiring many evaluations of $\tilde{x}_{0,\theta}$. Instead, like Campbell et al. [3] and Zhao et al. [26], we reverse many events at once. In particular we use an analogue of a k-Gillespie procedure [26]: we pick $k$ events to reverse, and then reverse them with a single evaluation of $\tilde{x}_{0,\theta}$.

To sample a point $x_0 \sim q(x_0)$ we first sample the noised sample $x_1 \sim q(x_1) = p(x_\infty)$ and the number of events in each dimension $S \sim q(S) = p(S)$. We now sample given a budget of $C$ evaluations of $\tilde{x}_{0,\theta}$. Every step we denoise the last $\lceil s_1/C \rceil$ events that have yet to be denoised. To denoise we can use Eqn. 20. In the case that we denoise $k \geq 1$ events for a dimension $d$ at once, we can use the fact that

$$p(\mathrm{pr}^k(x_t^d) | x_t, s_t) = \sum_{x_0^d} p(\mathrm{pr}^k(x_t^d) | x_t^d, s_t^d, x_0^d) p(x_0^d | x_t, S)$$

$$= \sum_{x_0^d} p(x_t^d | \mathrm{pr}^k(x_t^d)) p(\mathrm{pr}^k(x_t^d) | s_t^d, x_0^d) \frac{p(x_0^d | x_t, s_t)}{p(x_t^d | s_t^d, x_0^d)}.$$

We can write

$$p(x_t^d | \mathrm{pr}^k(x_t^d)) = \mathrm{pr}^k(x_t^d)^T K^k x_t^d$$

$$p(\mathrm{pr}^k(x_t^d) | s_t^d, x_0^d) = x_0^{d,T} K^{s_t^d - k} \mathrm{pr}^k(x_t^d)$$

And we can approximate the fraction with $\tilde{x}_{0,\theta}$ just as in Eqn. 20. Thus we define

$$q_\theta(\mathrm{pr}^k(x_t^d) | x_t, s_t) = K^k x_t^d \circ K^{s_t^d - k, T} \tilde{x}_{0,\theta}. \tag{21}$$

The total procedure is summarized in Alg. 2.

## B.4 Choosing the rate

Our choice of $\beta_t$ describes how we compress the forward process running from time 0 to $\int_0^1 \beta_s ds$ into the interval $[0, 1]$. $\beta_t$ controls what times we sample when training the objective Eqn. 6 and $\int_0^1 \beta_s ds$ controls the convergence of $p(x_1)$ to $p(x_\infty)$. Austin et al. [2] suggest picking $\beta_t$ so that the mutual information between $x_0$ and $x_t$ decreases linearly to $\epsilon$ on the interval $[0, 1]$. For SCUD models, we pick $\beta_t$ so that the same is true when conditioning on the schedule: $E_{s_t} \mathrm{MI}(x_0, x_t | s_t)$ decreases linearly on the interval $[0, 1]$.

**Mutual information rate functions**  To choose the rate function $\beta_t$, Austin et al. [2] calculated the frequency of tokens in the training data $p_0(b)$ and then calculated the joint distribution of $x_0$ and a particle which has evolved according to $\mathcal{L}$ for time $\tau$ along one dimension –

$$p(x_0 = b, x_\tau = b') = p_0(b)(e^{\tau \mathcal{L}})_{b,b'}.$$

They calculate the mutual information function $\mathrm{MI}(\tau)$ of this joint distribution; the mutual information is normalized so $\mathrm{MI}(0) = 1$. They then pick $\beta_t$ so that evolving in the modulated process linearly decreases the mutual information from 1 to $\epsilon$ on the interval $[0, 1]$, i.e. $\mathrm{MI}(\int_0^t \beta_s ds) = 1 - (1 - \epsilon)t$. For clarity, we'll set $\mathrm{MI}(\int_0^t \beta_s ds) = 1 - t$ and look at the interval $[0, 1 - \epsilon]$ below.

**Implementation in continuous time**  The process in [2] has discrete time, so the integral over $\beta$ is a sum and each $\beta_t$ can be pre-calculated before training begins. When we implement continuous time discrete diffusion, we use a Newton root finder to calculate $\int_0^t \beta_s ds = \mathrm{MI}^{-1}(1 - t)$ and the implicit function theorem to calculate $\beta_t = \frac{d}{dt} \int_0^t \beta_s ds = 1/\left(\frac{d}{dt} \mathrm{MI}(\int_0^t \beta_s ds)\right)$.

**Algorithm 2** Efficient sampling from SCUD

---

**Input:** function evaluation budget $C$.
// Sample $x_1, s_1$
**for** $d = 1, \ldots, D$ **do**
    $x^d \sim p(x_\infty)$
    $s^d \sim p(s_1) = \mathrm{Pois}(\int_0^1 \beta_s ds)$
**end for**
$L \leftarrow \lceil \sum_{d=1}^D s^d / C \rceil$                                ▷ Number of events to denoise per step
**for** $c = 1, \ldots, C$ **do**
    // Decide which positions to denoise in this step
    $k \leftarrow \vec{0}$
    **for** $\ell = 1, \ldots, L$ **do**
        **if** $\sum_{d=1}^D (s^d - k^d) > 0$ **then**                ▷ If there are remaining events to reverse...
            $d \sim \mathrm{Categorical}\left( \frac{s - k}{\sum_{d=1}^D (s^d - k^d)} \right)$    ▷ ...sample uniformly from remaining events in $s$.
            $k^d \leftarrow k^d + 1$
        **end if**
    **end for**
    // Denoise $k^d$ steps at each dimension $d$
    Predict $\tilde{x}_{0,\theta}(x, s)$
    **for** $d = 1, \ldots, D$ **do**
        $x^d \sim q_\theta(\mathrm{pr}^{k^d}(x^d)|x, s)$                         ▷ use Eqn. 21
        $s^d \leftarrow s^d - k^d$
    **end for**
**end for**
**Return:** $x$

---

**Schedules for SCUD** For SCUD, we instead calculate the joint distribution between $x_0$ and the particle after $m$ events, $x_{t_m}$, along one dimension –

$$p(x_0 = b, x_{t_m} = b') = p_0(b)(K^m)_{b,b'}.$$

Calling the mutual information between these variables $\mathrm{MI}_m$ we choose $\beta_t$ so that $E_{s_t} \mathrm{MI}_{s_t} = 1 - t$ where $s_t \sim \mathrm{Pois}(r^* \int_0^t \beta_s ds / \gamma)$. Again we calculate these values using a Newton root finder and the implicit function theorem.

**Connection to classical discrete diffusion** With this choice, note as $\gamma \to 0$, for any $\tau$

$$\mathrm{MI}_{r^* \tau / \gamma} = \mathrm{MI}(p_0(b)((I + \gamma \mathcal{L}/r^*)^{r^* \tau / \gamma})_{b,b'}) \to \mathrm{MI}(p_0(b)(e^{\tau \mathcal{L}})_{b,b'}) = \mathrm{MI}(\tau).$$

Therefore, $E_{s_t} \mathrm{MI}_{s_t} \to \mathrm{MI}(\int_0^t \beta_s ds)$, so $\int_0^t \beta_s ds$ converges to the same value as in classical discrete diffusion.

**Connection to masking discrete diffusion** In this case, $x_{t_m}$ is uniform independent of $x_0$ for all $m \geq 1$ Therefore, $\mathrm{MI}_m = 0$ for all $m \geq 1$ and $E_{s_t} \mathrm{MI}_{s_t} = e^{-\int_0^t \beta_s ds} = \alpha_t$. Therefore, $\alpha_t = 1 - t$.

## C  Structured processes

In this section we will describe the structured continuous time Markov processes we used in Sec. 6. Our processes are inspired by those from Austin et al. [2] and Alamdari et al. [1]; however those works framed the process in discrete time in such a way that they are not related to any continuous time Markov model, requiring us to design new processes. Note also that those works modified their processes to ensure that the transition matrix at every time-point was doubly stochastic; this was so that all transition matrices would have the same stationary distribution – a uniform distribution. In our case, we are free to pick any $\mathcal{L}$ that converges to a stationary distribution, even if it is not uniform.

### C.1 Gaussian process for images

To include the bias that two pixel values $i \neq j$ are similar if $(i-j)^2$ is small, we set $\mathcal{L}_{i,j} = \exp(-200\frac{(i-j)^2}{B})$ the value 200 was chosen as it gave the best results in small scale experiments. We then set $\mathcal{L}_{i,i} = -\sum_{j\neq i}\mathcal{L}_{i,j}$.

### C.2 Nearest neighbour process for language

Our vocabulary in the language result was approximately 30'000 tokens from the Bert-base-uncased tokenizer [6]. It is prohibitively expensive to compute a $30'000 \times 30'000$ matrix $K$ to take matrix vector products during training. Instead, we pick a sparse $K$ built using the embeddings from Devlin et al. [6]; for the most frequent 1000 words (which make up 95% of tokens seen in the data) $i, j$ we computed their similarity as $v_i^T v_j$ where $v_i$ is the normalized embedding of word $i$. For each word we found the 10 nearest neighbours; we noticed restricting to the top 1000 words resulted in nearest enighbours which were much more semantically similar. We next set, for nearest neighbours,

$$\tilde{\mathcal{L}}_{i,j} = \exp(v_i^T v_j / 0.3).$$

We next normalized $\tilde{\mathcal{L}}$ so that the diagonal is $1$ – this ensures that every word has an identical transition rate, avoiding the case where a word never transitions because it has no nearby neighbours.

We noticed that it often took a long time for particles to reach a stationary distribution with this process, so we added occasional transitions across the nearest neighbour graph; we called $p$ the normalized frequencies of the top 1000 words in the data and define the uniform transition infinitesimal generator

$$\mathcal{L}_{\text{unif}} = \mathbb{1} \otimes p - I,$$

where $\mathbb{1}$ is the vector of all 1's; this transitions tokens to a random token based on the final token's frequency in the data. We combine our two processes by defining

$$\mathcal{L} = \tilde{\mathcal{L}} + 0.4 \times \mathcal{L}_{\text{unif}}$$

and normalizing so that the smallest value on the diagonal was $-1$. We do not store this matrix explicitly, and only perform matrix operations with sparse matrix products and multiplication with $\mathbb{1}$ or $p$.

For tokens outside of the most frequent 1000, we transition using $\mathcal{L}_{\text{unif}}$.

### C.3 BLOSUM process for protein

BLOSUM is a matrix that can be describes how often different amino acids are seen in the same position in related protein families [10]. The $i, j$ entry of the matrix is

$$B_{i,j} = 2\log\frac{P_{ij}}{P_i P_j}$$

where $P_{ij}$ is the probability of two related proteins having amino acids $i, j$ at the same position, and $P_i$ is the marginal probability. We build a stochastic process to emulate drawing a related protein, so we set

$$K_{i,j} = \exp(B_{i,j}/2) \times P_j = P_{j|i}.$$

There are other letters in our vocabulary for non-canonical amino acids and padding; for $i$ not one of the canonical 20 amino acids, we set $K_{i,j} = P_j$, so all transitions an only occur to a canonical amino acid. Finally we set $\mathcal{L} = K - I$ (Fig. 5).

## D Experimental details

In all cases we trained models on 2 $A100$ GPUs on an academic cluster.

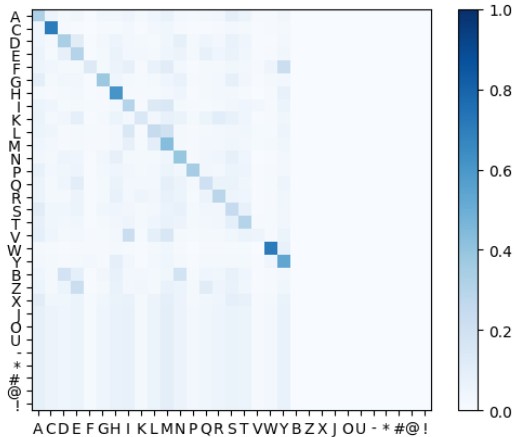

Figure 5: BLOSUM process $K$.

## D.1 Samples from image SCUD

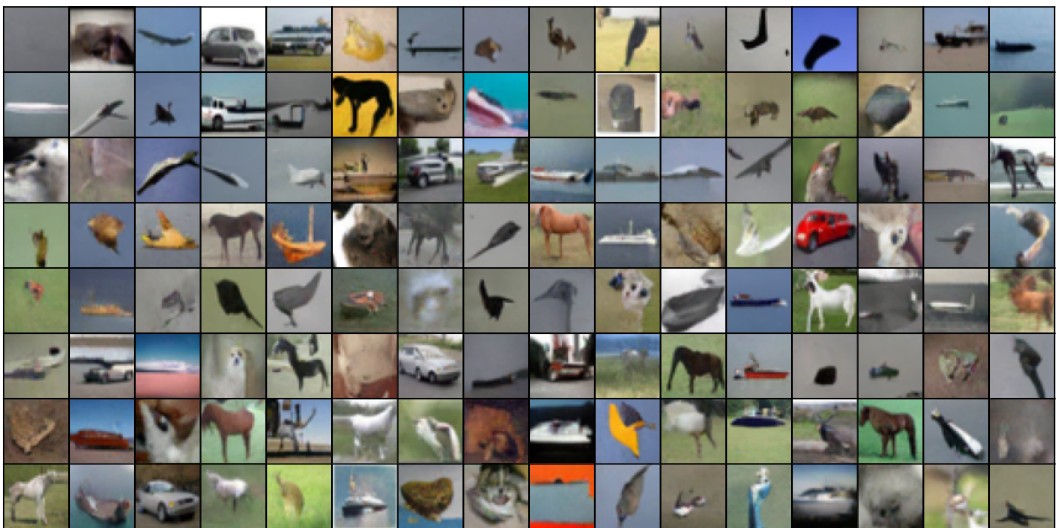

Figure 6: **Samples from SCUD Gaussian trained on CIFAR-10.**

## D.2 Pretrained protein weights

We train 150M parameter ESM2 models [12] with pre-trained weights on Uniref2 and compare to a masking diffusion model, DPLM [24], which used a similar strategy.

To get an ELBO for DPLM, we considered it as a discrete-time masking diffusion model with 500 timesteps. We used the formula from Austin et al. [2] to evaluate the ELBO of such a model. In particualr, the ELBO is

$$\sum_{t=1}^{500} \frac{1}{t} \mathbb{E}_{X_0, X_t} \sum_{i=1}^{L} \mathbb{1}(X_t^{(i)} = \text{mask}) \log q_\theta(X_0^{(i)} \mid X_t).$$

## D.3 Transition rates in Fig. 2

For $\tau$LDR we downloaded the CIFAR-10 model from `https://github.com/andrew-cr/tauLDR`. We simulated 2000000 forward trajectories using samples from CIFAR-10 and 1000 backward

| Method | Fwd. process | Perplexity |
|--------|--------------|------------|
| DPLM | Masking | 10.61 |
| Classical | BLOSUM | 10.34 |
| SCUD | Uniform | 10.95 |
| SCUD | BLOSUM | **10.04** |

Table 2: **Schedule conditioning improves uniform corruptions and accommodates structured forward processes for proteins, even when starting from pre-trained model weights.** We perform the protein experiment from the main text but replace the architecture with a pre-trained ESM2 backbone.

samples using $\tau$ leaping with $2^{16} = 65536$ steps. For forward samples $x$ we calculated rates $-\mathcal{L}_{x,x}$ and for backward samples we calculated rates $-\mathcal{L}_{\theta,x,x}$. We then averaged forward and backward rates at each timestep and calculate the difference between the average forward and backward rates.

For D3PM we download the 640M uniform model from `https://github.com/microsoft/evodiff`. We were able to calculate the forward rates analytically. We simulated 1000 backward samples of length 110 and at each time step we calculated the probability of a transition; we multiplied this probability by $1/\Delta t$ to get a rate. We did not average with a sliding window.

### D.4   Images

We use an architecture inspired by Kingma et al. [11] like in MD4 [23] with a slight modification to incorporate $s_t$. The architecture first embeds $x_0$ like in Shi et al. [23] and then puts it through a UNet with 32 layers and no up- or down-sampling. At every layer of the UNet, a feed forward layer is applied to a sinusoidal embedding of the time $t$ and the output is added to the channels at every pixel – ax position $i, j$, activations $a^{i,j}$ at updated

$$a^{i,j} \leftarrow \mathrm{FF}_\theta(\mathrm{emb}(t)) + a^{i,j}.$$

Instead each activation is updated using a FiLM layer using the number of events up to time $t$.

$$a^{i,j} \leftarrow \mathrm{FF}_{1,\theta}(\mathrm{emb}(s_t^{i,j})) + \mathrm{FF}_{2,\theta}(\mathrm{emb}(s_t^{i,j})) \circ a^{i,j}.$$

The feed forward layers are shared across every position $i, j$. We used the same training parameters as in Shi et al. [23]; we trained each of our large models for 2 days and each of our models from Fig. 1 took between 1.5 and 2 hours. Our large models were trained on $2.6 \times 10^9$ images, the same amount as MD4 [23] and $\tau$LDR [3] and compared to $1.9 \times 10^9$ for D3PM [2].

We use $K = 2048$ function evaluations to generate images. The results of Fig. 1 used a batch size of 16 and the same architecture but with an 8 layer UNet – masking and classical models used FiLM layers with $t$ instead of $s_t$.

### D.5   Language

We use the diffusion transformer architecture [17] as in SEDD [15]. This architecture has FiLM layers to add $t$ at each layer; as above, we replace $t$ with $s_t$. We use the training settings as in SEDD [15], accumulating to match their batch size of 512. We trained our models for 2 days each on $1.1 \times 10^{10}$ tokens.

### D.6   Protein

We use the small CARP architecture from [1]. The original architecture added as embedding of $t$ at the first layer. We add FiLM layers for $s_t$ at every layer as described above. We train and test on the March 2020 release of Uniprot2020 released by Alamdari et al. [1]. We use a batch size of 128 protein up to size 1024 as in Alamdari et al. [1], randomly truncating proteins over that size. We trained each model for 2 days on $1.1 \times 10^{11}$ tokens, compared with $3.3 \times 10^{11}$ for the baseline D3PM models from Alamdari et al. [1].

### D.7   Computational complexity

In terms of computational complexity, the major differences between SCUD and classical discrete diffusion are (A) replacing operations of $\mathcal{L}$ with operations of $K$, and (B) replacing the time $t$ in the

argument of $\tilde{x}_{0,\theta}$ with the number of transitions $S$. We discuss how (B) does not result in a large increase of computational complexity below, and note that (A) does not change the computational complexity except when the number of tokens $B$ is large, when it actually enables strategies that reduce complexity.

**(A) Matrix computations** To calculate our loss, Eqn. 6, in Eqn. 6 we see that we only need to take matrix vector products with K; the analogous quantity in classical discrete diffusion requires matrix exponentiation $\exp(t\mathcal{L})$ (Luo et al., 2022). When $B$ is small, both these calculations have negligible complexity and can be calculated similarly quickly by precomputing an eigen-decomposition of $K$ or $\mathcal{L}$. But when $B$ is large, as in the language modeling case, these calculations become very expensive; Luo et al., 2022 settled for very simple $\mathcal{L}$, masking and uniform, such that $\exp(t\mathcal{L})$ can be easily analytically calculated; SCUD is able to build in a richer forward process by picking a sparse + low rank $K$ so that matrix vector products are very fast.

In terms of big-O notation, when an eigendecomposition is precomputed, $(\exp(t\mathcal{L})\tilde{x}_0^d)_{d=1}^D$ and $(K^{s_t^d}\tilde{x}_0^d)_{d=1}^D$ each cost $\Theta(DB^2)$ for two dense matrix multiplies and a scaling by the exponentiation or power of the eigenvalues. When $K^{s_t^d}$ is a sparse matrix with $O(rB)$ entries or has a rank of $r$, calculating $(K^{s_t^d}\tilde{x}_0^d)_{d=1}^D$ is $O(DBr\max_d(s_t^d))$; in our language case, $B$ is large while $\max_d(s_t^d) \approx 30$ and we pick $r \approx 20$ resulting in a large speedup.

**(B) Computations with $S$** Indeed, the place that SCUD adds some overhead to calculations is in replacing the arguments of $\tilde{x}_{0,\theta}(x_t, \cdot)$: the time over which $x_0$ has been corrupted, $t$, a scalar, is replaced with the number of corruptions of each token $S$, a $D$-dimensional object. The overhead of this operation is dependent on the architecture of $\tilde{x}_{0,\theta}$. We picked $\tilde{x}_{0,\theta}$ so that no parameters were added by replacing $t$ with $S$, and such that the computational and memory overhead caused by this replacement were negligible compared to the operations and memory spent on operations on the $D$-dimensional $x_t$. Above we used previous architectures modified so that each operation on $t$ was also applied to each dimension of $S$. As well, for the architectures we chose, whenever a function of $t$ was added or multiplied to a set of activations, say at layer $\ell$, $h_{\ell,\theta}$, the activations had a dimension $D$, so we could perform the same operation with element-wise addition or multiplication with $S$, i.e.

$$h_{\ell+1,\theta}^d = f_{1,\theta}(t)h_{\ell,\theta}^d + f_{2,\theta}(t) \text{ was replaced with } h_{\ell+1,\theta}^d = f_{1,\theta}(s_t^d)h_{\ell,\theta}^d + f_{2,\theta}(s_t^d).$$

Thus, adapting $\tilde{x}_{0,\theta}$ for SCUD in this way adds no extra parameters. The overhead of this change is that every call to $f_\theta$ is replaced by $D$ calls, $D$-times the activations $f_\theta(s_t^d)$ must be stored, and $D$-times more gradients must be calculated for $f_\theta(s_t^d)$. $f_\theta$ is however a set of linear functions and activations. The operations on the corrupted data $x_t$ involve convolutions and attention, which have much larger memory and computational costs. In big-O notation, the cost of calculating $\tilde{x}_{0,\theta}(x_t, t)$ and $\tilde{x}_{0,\theta}(x_t, S)$ are therefore identical – at worst, the constant in front of the largest term changes. Therefore, in our experiments, we ran all models for roughly equal time with the same batch sizes and did not observe any substantial difference in computation.

## E  Extending flow matching to SCUD

Here we follow the exposition of Campbell et al. [4] to derive flow matching models that are conditioned on schedule. In App. E.1 we derive schedule conditioned flow matching (SCUM) in generality. In App. E.2 we describe how SCUD is an instance of SCUM and show how by training a SCUD model, one can sample from a large class of SCUM models. Finally in App. E.3 we derive an example class of SCUM models. The conclusion is that schedule conditioning can be extended to the flow matching case just as classical discrete diffusion can.

### E.1  Schedule conditioned flow matching (SCUM)

We consider discrete objects in a set of size $B$ and in this quick exposition leave out the multi-dimensional case as an easy extension of the logic of SCUD or Campbell et al. [4]. In flow matching, we wish to approximately sample from a target $p(x_0)$ (this is called $x_1$ in Campbell et al. [4]). In regular flow matching, we define distributions of samples noised for time $t$: $p(x_t|x_1)$ (Eqn. 6 of Campbell et al. [4]). To condition on the schedule, we instead define distributions of samples that have been noised by $s$ events from $x_1$: $p(x_s|x_0)$. We assume $p(x_s|x_1)$ is close to an easy to sample

from distribution $p(x_\infty)$ when $s$ has large entries. In particular, for $s$ with large entries, the marginal $p(x_s) \approx p(x_\infty)$; Now we want to denoise events to get $p(x_{s-1})$ and ultimately $p(x_0)$ (Eqn. 5 of Campbell et al. [4]).

To do so, we first choose how to denoise elements in $p(x_s|x_0)$. Say $K_{s|x_0}$ is a stochastic matrix such that sampling $p(x_s|x_0)$ then $x_{s-1} \sim \mathrm{Categorical}(K^T_{s,d|x_0} x^d_s)$ gives a sample from $p(x_{s-1}|x_0)$. The next result is the analogous result of Prop. 3.1 of Campbell et al. [4]: given a sample from the marginal, $x_s \sim p(x_s)$ we can denoise an event in dimension $d$ by averaging over $x_0|x_s$ and using $K_{s|x_0}$.

**Proposition E.1.** *Define* $K_{s;x_s,\cdot} = E_{p(x_0|x_s)} K_{s|x_0;x_s,\cdot}$. *Then sampling* $x_s \sim p(x_s)$ *and* $x_{s-1} \sim$ $\mathrm{Categorical}(K_{s;x_s,\cdot})$ *gives a sample from* $p(x_{s-1})$.

*Proof.*

$$E_{p(x_s)} E_{p(x|x_s)} K_{s|x_0;x_s,x_{s-1}} = \sum_{x_0} p(x_0) \left( E_{p(x_s|x_0)} K_{s|x_0;x_s,x_{s-1}} \right)$$

$$= \sum_{x_0} p(x_0) p(x_{s-1}|x)$$

$$= p(x_{s-1}).$$

$\square$

Given this result, we can define schedule conditioned flow matching models (SCUM). First we approximate $p(x_0|x_s)$ with a neural network $\tilde{x}_{0,\theta}(x_s, s)$; next we sample from $p(x_\infty)$ which is $\approx p(x_s)$ for some large $s$, and then iteratively denoise by approximating $K_{s;x_s,\cdot}$. (Alg. 3).

---

**Algorithm 3** Sampling from SCUM in analogy to Alg. 1 in Campbell et al. [4]

$s \leftarrow$ large number
$x_s \sim p(x_\infty) \approx p(x_s)$
**while** $s > 0$ **do**
$\quad K_{s;x_s,\cdot} \leftarrow E_{\tilde{x}_{0,\theta}(x_s,s)} K_{s|x_0;x_s,\cdot}$
$\quad x_{s-1} \sim \mathrm{Categorical}(K_{s;x_s,\cdot})$
$\quad s \leftarrow s - 1$
**end while**
**Return:** $x_0$

---

To train $\tilde{x}_{0,\theta}(x_s, s)$ we can just minimize the cross entropy

$$E_{s \sim \mathrm{Unif}(1,2,\ldots,\text{large number}), p(x_0), p(x_s|x_0)} x_0^T \log \tilde{x}_{0,\theta}(x_s, s).$$

We could alternatively use a different distribution for $s$, such as a Poisson. Note that $p(x_s|x_0)$ does not depend on the particular choice of $K_{s|x_0}$, so we can train $\tilde{x}_{0,\theta}(x_s, s)$ once and then decide the best $K_{s|x_0}$ for sampling at test time.

### E.2 SCUD is SCUM

We now show that for a particular choice of $K_{s|x_0}$, the simulated trajectories of SCUM are that of SCUD as in Appendix H of Campbell et al. [4]. Next we discuss how, given a trained SCUD model we can sample from a wide variety of SCUM models.

Define a Markov process that noises datapoints $x_0$ with an infinitesimal generator $\mathcal{L}$ with rate function $\beta_t$. Say we have a data point $x_0$ that's been noised $s > 0$ times and define $K_{s|x_0;x_s,\cdot} = p(\mathrm{pr}(x_s)|x_s, x_0, s)$ as in Eqn. 5. Then

$$K_{s;x_s,\cdot} = E_{p(x_0|x_s)} K_{s|x_0;x_s,\cdot}$$

$$= E_{p(x_0|x_s,s)} p(\mathrm{pr}(x_s)|x_s, x_0, s)$$

$$= p(\mathrm{pr}(x_s)|x_s, s)$$

which is exactly the distribution we approximate to denoise an event in SCUD (Alg. 2). Therefore SCUD is just SCUM with a particular choice of $K_{s|x_0}$, with "large number" in Alg. 3 set to $\mathrm{Pois}(\int_0^t \beta_s ds)$.

Furthermore, SCUD trains a $\tilde{x}_{0,\theta}(x_s, s)$ to predict $x_0$ given $x_s, s^2$. Campbell et al. [4] suggests that an advantage of flow matching is that one can train $\tilde{x}_{0,\theta}$ once and then decide on the best infinitesimal generator at test time; we can do the same by training $\tilde{x}_{0,\theta}$ with the SCUD objective and then changing $K_{s|x_0}$ at test time.

### E.3 Examples of SCUM

Say we have built a SCUD model with transition matrix $K$. The canonical choice for $K_{s|x_0}$ above is

$$K_{s|x_0;x_s,x_{s-1}} = x_{s-1}^T K x_s \frac{x_0^T K^{s-1} x_{s-1}}{x_0^T K^s x_s}.$$

We now describe a family of $K_{s|x_0}$ that can be alternatively used to sample from $p(x_0)$.

First note that for SCUD, $p(x_s|x_0) = K^{s,T} x_0$. Therefore $K_{s|x_0}$ can be any matrix with $K_{x|x_0}^T K^{s,T} x_0 = K^{s-1,T} x_0$ and positive entries with rows that add to 1. Campbell et al. [4] suggested picking the process to minimally move mass from position with too much in $K^{s-1,T} x_0$ to those with too little in $K^{s,T} x_0$ ($R^*$ in Prop. 3.2 in Campbell et al. [4]); we can do that with the choice

$$K_{s|x_0;x_s,y}^* = \frac{\mathrm{ReLU}(y^T K^{s-1,T} x_0 - y^T K^{s,T} x_0)}{\sum_z \mathrm{ReLU}(z^T K^{s-1,T} x_0 - z^T K^{s,T} x_0)} \times \mathrm{ReLU}(x_s^T K^{s,T} x_0 - x_s^T K^{s-1,T} x_0)$$

for $x_s \neq y$, which moves mass from $x_s$ with too much mass to $y$ with too little in proportion to how much mass they need.

To augment this "most efficient" choice Campbell et al. [4] describe a method to add stochasticity to $K_{s|x_0}$. They do so by introducing an infinitesimal generator that obeys details balance; we do the same. Say $\mathcal{L}_{s|x_0}^{\mathrm{DB}}$ keeps the distribution $p(x_s|x_0)$ stationary, say by satisfying detailed balance. Then we can add more noise to $K_{s|x_0;x_s,y}$ by defining $K_{s|x_0}^\eta = e^{\eta \mathcal{L}^{\mathrm{DB}}} K_{s|x_0}^*$ since

$$K_{x|x_0}^T K^{s,T} x_0 = K_{s|x_0}^* e^{\eta \mathcal{L}^{\mathrm{DB}}} K^{s,T} x_0 = K_{s|x_0}^* K^{s,T} x_0 = K^{s-1,T} x_0.$$

By varying $\eta$, Campbell et al. [4] optimized samples for stochasticity against likelihood.

In conclusion, just as one can do with classical discrete diffusion models, after training a SCUD model, one can optimize a stochasticity parameter $\eta$ to get desirable samples.

---

[2] In the high dimensional case, unlike our exposition of SCUM, SCUD trains $\tilde{x}_{0,\theta}(x_s, s)$ to approximate, for each dimension $d$, $p(x_0^d|x_s^{-d}, s)$ rather than $p(x_0^d|x_s, s)$ (Sec. B.2). However any prediction of $p(x_0^d|x_s^{-d}, s)$ can be transformed into a prediction of $p(x_0^d|x_s, s)$ via the identity $p(x_0^d|x_s, s) \propto p(x_s^d|s^d, x_0^d) p(x_0^d|x_s, s)$ which doesn't depend on the specific choice of $K_{s|x_0}$ – the difference is just a matter of parameterization.

