# OpenReview forum: "Why Masking Diffusion Works: Condition on the Jump Schedule for Improved Discrete Diffusion"
_NeurIPS.cc/2025/Conference — NeurIPS 2025 poster_

### Official Review · Reviewer_zwCo · 2025-06-29

**Clarity:** 1
**Significance:** 3
**Originality:** 3
**Rating:** 3
**Confidence:** 4

**Summary:**

This paper introduces Scheduled Conditioned Diffusion (SCUD), a generative modeling framework that first samples a transition schedule and then generates data conditioned on this sampled schedule. The approach aims to generalize discrete diffusion by decoupling the temporal structure from the transition dynamics. While the core idea of conditioning on a sampled schedule is novel and has potential, the current version of the paper lacks clarity in its mathematical formulation, contains questionable assumptions (particularly regarding the transition process and its independence from the schedule), and requires stronger theoretical and empirical justification. As it stands, the work is promising but not yet ready for publication.

**Questions:**

Major Questions:
- See weaknesses

Minor Question
- Why the shorthand for Scheduled conditioned diffusion written as "SCUD"?

**Ethical Concerns:**

["NO or VERY MINOR ethics concerns only"]

**Final Justification:**

Thank you for the detailed response. The clarification on transitions vs. events resolved several of my concerns. I now better appreciate the theoretical framework and am raising my score accordingly.

**Limitations:**

yes

**Quality:**

2

**Strengths And Weaknesses:**

Strengths

The idea of conditioning the generative process on a sampled schedule of transitions is novel and conceptually intriguing.
It offers a new perspective on incorporating temporal flexibility into discrete diffusion frameworks.


Weaknesses
1. Conceptual and Mathematical Concerns Regarding the Transition Matrix $K$
The definition and usage of matrix $K$ (as described in lines 129-136 and Proposition 4.1) raise several conceptual issues:
- Once $\Delta t$ is sampled, the first transition occurs at time $t+\Delta t$, so it is unclear why $x_t$ and $x_{t+\Delta t}$ could be the same.
- The proof of Proposition 4.1 appears to implicitly assume that $\Delta t$ is infinitesimally small, but this assumption is neither stated nor justified.
- When sampling $x_{t+\Delta t}$ from a categorical distribution parameterized by $K_{x_t}$, it is unclear why this process would be independent of the sampled $\Delta t$, especially when $\Delta t$ governs the time horizon of transitions.

2. Incorrect Claim About Equivalence with Uniform Transition
- The statement in lines 205–209 is misleading. Even under uniform transition schedules, transition and masking processes are not equivalent, even if the transition schedule is known. This is because under uniform transitions, the same token position can undergo multiple transitions, which breaks the claimed equivalence with masking.

3. Notation and Mathematical Clarity
Several mathematical definitions and notations are unclear or inconsistent:
- In Figure 3, the role of $\tilde{x}_{0, \theta}$  is ambiguous. If this represents a model prediction, is it a denoised sample or a score? In discrete diffusion, the model typically estimates log-ratio scores [1].
- In Equation (5), it is not clear whether the schedule variable $s_t$ is discrete or continuous. Since it's drawn from an exponential distribution, one would expect it to be continuous.
- In Proposition 4.4, it appears the authors directly model $q_{\theta}$ . However, prior work [1] emphasizes estimating the log-ratio between adjacent states, which is not discussed here.
- The notation $\tilde{x}$ is used without definition in Proposition 5.1.
- Proposition 5.2 lacks context and should clearly describe the connection to SEDD loss. The statement about equivalence needs to be explained more rigorously.

4. Experimental Evaluation
- The empirical validation could be strengthened by including comparisons against more competitive or recent baselines such as SEDD or LLADA, which are more aligned with the proposed approach in terms of model class and task setup.

[1] Lou, Aaron, Chenlin Meng, and Stefano Ermon. "Discrete Diffusion Modeling by Estimating the Ratios of the Data Distribution." ICML 2024.

---

> ### Author Rebuttal · Authors · 2025-07-30
>
> Thank you for your thoughtful and effortful review! We clarify our theoretical and experimental results below. However, we note your appraisal about clarity differs from the other reviewers and your questions concern the foundations of continuous-time Markov chains. We recognize that readers would benefit from some familiarity with continuous-time Markov chains. We'll add additional mathematical background to the appendix and note Chapter 2 of Liggett 2010 and the τLDR paper provide excellent primers in this established area.
>
> > Conceptual and Mathematical Concerns Regarding the Transition Matrix $K$ The definition and usage of matrix $K$ (as described in lines 129-136 and Proposition 4.1) raise several conceptual issues:
>
> > Once $\Delta t$ is sampled, the first transition occurs at time $t+\Delta t$, so it is unclear why $x_t$ and $x_{t+\Delta t}$ could be the same.
>
> Our definition and usage of the matrix $K$, including self-transitions, is standard (ex Chapter 2 of Liggett 2010). In our first discussion of the Gillespie algorithm we defined $\Delta t$ as the transition times. However, Prop 4.1 shows that we can simulate from a Markov process defined by $\mathcal L$ even if there isn’t a transition at each $\Delta t$. This is important for applying SCUD to processes which have $\mathcal L$ with non-constant diagonals. The beginning of Sec 5 describes how decreasing the probability of $x_t=x_{t+\Delta t}$ leads to more schedule conditioning.
>
> > The proof of Proposition 4.1 appears to implicitly assume that $\Delta t$ is infinitesimally small, but this assumption is neither stated nor justified.
>
> Prop 4.1 simply uses the formal definition of the infinitesimal generator $\mathcal L$ (Equation 2.4 of Liggett 2010) – the limit $t\to 0$ in not an assumption about the time between transitions. Note we are working in continuous time, so there is no minimum $\Delta t$ and the markov process can be uniquely described by a matrix $\mathcal L$ which represents the infinitesimal flow at a particular time. The formal connection between $\mathcal L$ and the Markov process is described in depth in chapter 2 of Liggett 2010.
>
> > When sampling $x_{t+\Delta t}$ from a categorical distribution parameterized by $K_{x_t’}$, it is unclear why this process would be independent of the sampled $\Delta t$, especially when $\Delta t$ governs the time horizon of transitions.
>
> This is a property common to all Markov processes. The “event” times are drawn from exponential distributions which are “memory-less”, and then $K$ describes the behaviour at each event. The sampling algorithm we describe is known as the Gillespie algorithm.
>
>
> > Incorrect Claim About Equivalence with Uniform Transition: The statement in lines 205–209 is misleading. Even under uniform transition schedules, transition and masking processes are not equivalent, even if the transition schedule is known. This is because under uniform transitions, the same token position can undergo multiple transitions, which breaks the claimed equivalence with masking.
>
> Lines 205-209 give an informal argument and the formal argument is contained in Proposition 5.1.
>
> Intuitively, the reason the “multiple transitions” doesn’t represent a meaningful difference between the two classes of models is that a position that has transitioned uniformly just once has lost all of its information – therefore when trying to predict $x_0$, knowing that a token in $x_t$ has been transitioned once or many times presents the same information.
>
> > Notation and Mathematical Clarity Several mathematical definitions and notations are unclear or inconsistent:
>
> > In Figure 3, the role of  is ambiguous. If this represents a model prediction, is it a denoised sample or a score? In discrete diffusion, the model typically estimates log-ratio scores [1].
>
> > In Proposition 4.4, it appears the authors directly model $x_0$. However, prior work [1] emphasizes estimating the log-ratio between adjacent states, which is not discussed here.
>
> Figure 3 represents predicting the “denoised” letter $x_0$. We will make this more clear in Section 4.2. The parameterization we use is the most common (used by continuous diffusion models as well as D3PM, EvoDiff, and masking diffusion models). However we are familiar with the SEDD (your citation [1]) parameterization and, in fact, one of our central results, Proposition 5.2, actually describes the connection to our parameterization; it shows that the two parameterizations are equivalent.
>
> > In Equation (5), it is not clear whether the schedule variable $s$ is discrete or continuous. Since it's drawn from an exponential distribution, one would expect it to be continuous.
>
> Line 155 defines $s_t$ as the number of events that have occurred so far, not the event times $S$. It is distributed as a Poisson as described in Algorithm 2 for example. We will make this more clear.
>
> > The notation $\tilde x$ is used without definition in Proposition 5.1.
>
> $\tilde x_{0, \theta}$ is the prediction of the denoised sequence. We will restate the definition in Sec 5 for clarity.
>
> > Proposition 5.2 lacks context and should clearly describe the connection to SEDD loss. The statement about equivalence needs to be explained more rigorously.
>
> Proposition 5.2 is formally stated and proved in the appendix; there are links in the pdf between the statement in the main text and the proof. However the SEDD loss is cumbersome to write, so we simply moved its statement to the appendix.
>
> > The empirical validation could be strengthened by including comparisons against more competitive or recent baselines such as SEDD or LLADA, which are more aligned with the proposed approach in terms of model class and task setup.
>
> Our experiments aimed to confirm our theoretical results in practice and leverage them to interrogate some commonly held beliefs around discrete diffusion models. Nevertheless, while not necessary for proving our main claims, our implementation of our SCUD uniform uses the SEDD architecture and can therefore be interpreted as a comparison to SEDD.
>
> In brief, to compliment our theoretical contributions, our goal was to demonstrate our theoretical results in practice, and leverage the theory to reinvestigate structured processes in the literature. In the case of D3PM’s Gaussian and EvoDiff’s BLOSUM, SCUD leads to state-of-the art results, but for D3PM’s sparse graph process SCUD leads to less of an improvement. We will make our goal clearer in future drafts.
>
> In particular, our goal in the experiments section was to
>
> 1. Show that schedule conditioning improves performance. This was validated by comparison to previous sota models in theory, images, and proteins. Technical considerations meant we were limited to uniform $\mathcal L$, where the superiority of masking was already established.
>
> 2. Redo previous experiments that showed that structured processes performed worse than masking. Our theory suggested these were unfair – masking had the advantage of schedule conditioning while structured processes did not. We redid these fairly by building SCUD versions of structured models. But we are not necessarily advocating for using these specific structured processes. And we of course do not expect every structured forward process will lead to an improvement in performance.
>
> For images and proteins, we saw that Gaussian and BLOSUM processes improve the fit to the data, achieving state-of-the-art fit. For language models, we saw a very small improvement from the graph structure suggested in the D3PM paper, potentially suggesting there is room to see the sorts of improvements from BLOSUM and Gaussian processes but in language by building an improved model.
>
> Furthermore, our models trained on LM1B were trained using the SEDD architecture, so we expect SCUD uniform to be nearly identical to SEDD masking. Therefore one could consider our results a comparison of SEDD / SCUD masking and SCUD structured in both runtime and performance. Finally, models in MDLM and SEDD were trained longer on LM1B than our models, so their performances could not be directly compared. We reasoned that training longer to compare with those models would not impact our claims (1) or (2) since the structured process performs so similarly to SEDD, so we did not do this comparison. Instead we took the language example as an opportunity to demonstrate the scaling of SCUD to huge $K$. This opens the door to more creative forward processes for language making use of other structured $K$.
>
> Finally, we compare to more recent discrete diffusion models in the protein case in our response to Reviewer hLSS (search for "New state-of-the-art protein results on DPLM").

---

> > ### Comment · Reviewer_zwCo · 2025-08-06
> >
> > Thank you for the detailed response. The clarification regarding transitions vs. events addressed several of my concerns, and I now have a clearer understanding of the theoretical results. As a result, I have raised my score.
> >
> > That said, I still believe the paper could be improved in terms of presentation. The notation and exposition remain somewhat challenging to follow (at least for me), and key definitions are often difficult to locate when needed. Enhancing the structure and organization of the paper would significantly improve its clarity and accessibility.

---

> > > ### Author Response · Authors · 2025-08-06
> > >
> > > Thank you for your thoughtful response and raising your score! To summarize our promised changes for the record:
> > >
> > > * We add a definitions box to Section 2 describing $\mathcal L, K, S, t, (x_t)_t$ and Gillespe's algorithm, with citations to good primers on continuous time Markov processes.
> > >
> > > * We add a "re-definitions" box to Section 4.1 describing that we now allow self-transitions in $K$, and re-define $S, s_t$ as "event" times rather than transition times. We clarify the difference between $S$ and $s_t$.
> > >
> > > * We add a definitions box to Section 4.2 describing $\tilde x_{0, \theta}$ and comparing with other common parameterizations.
> > >
> > > We appreciate your feedback!

---

### Official Review · Reviewer_hLSS · 2025-07-02

**Clarity:** 4
**Significance:** 3
**Originality:** 4
**Rating:** 4
**Confidence:** 4

**Summary:**

This paper aims to answer why masked discrete diffusion works so much better than prior discrete diffusion approaches.  It finds that the main cause of this is that the forward and reverse processes induced by masked discrete diffusion have the property that the masked diffusion reverse process inherently matches the forward process in terms of *when* a state should transition.  This is in contrast to other discrete diffusion processes where the learned reverse process may induce some error due to a mismatch in the transition times (as represented in the paper by a novel decomposition of the ELBO).  The authors show that by explicitly conditioning other classical discrete diffusion processes on an event schedule they are able to match or exceed the performance of masked diffusion models on a variety of tasks.

**Questions:**

1. How does your protein performance compare to current state of the art models DPLM/DPLM2?

2. How well does SCUD perform on unconditional protein generation?

3. How does SCUD compare to MDLM on LM1B (or MDLM/MD4 on OpenWebText)?

**Ethical Concerns:**

["NO or VERY MINOR ethics concerns only"]

**Final Justification:**

I found the paper to be well written and strongly motivated.  While I would have preferred to recommend a full acceptance, I chose borderline accept as I believe the experimental results were not as strong as I would have liked -- the image results were good, but the protein results while improving on baseline (which was understandably low due to experimental setup) only achieved a relatively low 47 pLDDT, and it would be nice to show more convincing proof of the results on text.  However, the theoretical contribution and manuscript quality is I believe significant enough to recommend a 4.

**Limitations:**

Yes

**Quality:**

3

**Strengths And Weaknesses:**

I very much enjoyed reading this paper -- I found it to be well written and clear and that its theoretical contributions were both novel and significant.  However, I have some concerns regarding the empirical evaluation of SCUD.  Specifically, I found that the validation for both proteins and text to be rather lacking (though the image results were quite good).

For proteins the paper does not compare to the current state-of-the-art masked diffusion model for proteins in DPLM/DPLM2 [1, 2] which degrades the experimental quality.  It would also be nice to see how SCUD performs on unconditional generation as this is one of the main metrics one looks for to assess protein language models (see [1,2,3] for examples of these metrics).

With regards to text, the test perplexities on LM1B are significantly higher than those reported in MDLM [4].  I realize that MDLM was trained on more tokens than the SCUD models in the paper, but the paper's main claims would be significantly more compelling if it is shown that SCUD improves non-masked diffusion performance to comparable levels of performance on text as in MDLM or MD4 [5].

Overall, I am currently leaning slightly towards rejection on this paper.  However, this is only because I feel the empirical validation is still a bit lacking in the current manuscript.  I believe the theoretical contributions in the paper are significant and I would be more than happy to increase my score if the authors can provide additional experimental results along the lines of the concerns I listed above.  Finally, I note that I am most curious about the text experiments and would weight these slightly higher than proteins.

References:

[1] Wang, Xinyou, et al. "Diffusion language models are versatile protein learners." arXiv (2024).

[2] Wang, Xinyou, et al. "Dplm-2: A multimodal diffusion protein language model." arXiv (2024).

[3] Hayes, Thomas, et al. "Simulating 500 million years of evolution with a language model." Science (2025).

[4] Sahoo, Subham, et al. "Simple and effective masked diffusion language models." Advances in Neural Information Processing Systems 37 (2024).

[5] Shi, Jiaxin, et al. "Simplified and generalized masked diffusion for discrete data." Advances in neural information processing systems 37 (2024).

---

> ### Author Rebuttal · Authors · 2025-07-30
>
> Thank you for your thoughtful review! We are glad you appreciated the theoretical content and address your concerns about the experiments below with a more in depth experiment on using pre-trained weights and DPLM. With this result, we achieve state-of-the-art on both images and proteins.
>
> First let us recap the goals of our experiments to describe how SCUD compares to MDLM and SEDD and why we chose to do further experiments on DPLM.
>
> In brief, to compliment our theoretical contributions, our goal was to demonstrate our theoretical results in practice, and leverage the theory to reinvestigate structured processes in the literature. In the case of D3PM’s Gaussian and EvoDiff’s BLOSUM, SCUD leads to state-of-the art results, but for D3PM’s sparse graph process SCUD leads to less of an improvement. We will make our goal clearer in future drafts.
>
> In particular, our goal in the experiments section was to
>
> 1. Show that schedule conditioning improves performance. This was validated by comparison to previous sota models in theory, images, and proteins. Technical considerations meant we were limited to uniform $\mathcal L$, where the superiority of masking was already established.
>
> 2. Redo previous experiments that showed that structured processes performed worse than masking. Our theory suggested these were unfair – masking had the advantage of schedule conditioning while structured processes did not. We redid these fairly by building SCUD versions of structured models. But we are not necessarily advocating for using these specific structured processes. And we of course do not expect every structured forward process will lead to an improvement in performance.
>
> For images and proteins, we saw that Gaussian and BLOSUM processes improve the fit to the data, achieving state-of-the-art fit. For language models, we saw a very small improvement from the graph structure suggested in the D3PM paper, potentially suggesting there is room to see the sorts of improvements from BLOSUM and Gaussian processes but in language by building an improved model.
>
> Furthermore, our models trained on LM1B were trained using the SEDD architecture, so we expect SCUD uniform to be nearly identical to SEDD masking. Therefore one could consider our results a comparison of SEDD / SCUD masking and SCUD structured in both runtime and performance. Finally, as you mentioned, models in MDLM and SEDD were trained longer on LM1B than our models, so their performances could not be directly compared. We reasoned that training longer to compare with those models would not impact our claims (1) or (2) since the structured process performs so similarly to SEDD, so we did not do this comparison.
>
> **New state-of-the-art protein results on DPLM**
>
> Since 1) the EvoDiff’s BLOSUM process lead to a greater boost in performance than DPLM’s language graph process; and 2) DPLM starts with pretrained weights from ESM3 making it an interesting test case to show that SCUD can also leverage pre-trained weights; we elected to do further experiments in the protein setting. Note however that DPLM was trained with a reweighted ELBO as an objective, and developed a specialized sampling procedure that does not sample from the learned distribution; therefore, its performance as an unconditional generative model, which is what we are interested in this work, may not reflect the performance on design tasks the authors demonstrated in their paper. Below we report DPLM perplexities and sample quality according to how DPLM was trained – as a discrete-time masking diffusion model with $\Delta t=1/500$.
>
> In new experiments, we started with DPLM medium architectures, added FiLM layers to the pretrained ESM models and trained with BLOSUM classical, uniform SCUD / masking and BLOSUM SCUD. During the rebuttal period we managed to train on 4.3 million tokens; unfortunately this is less than 20% of the training budget from DPLM. Nevertheless, we show that SCUD BLOSUM indeed achieves the best likelihoods. For sample sample quality we measured by mean predicted foldability (pLDDT) of 100 sequences of length 200) and used 500 model evaluations. We see that SCUD is the best among our model implementations and is competitive with DPLM. Note DPLM was trained for much longer, which is known to increase the confidence of model predictions – indeed the sample quality of our models are still improving. We will update these numbers for models trained for longer in the final draft.
> |Model | perplexities | mean pLDDT from OmegaFold |
> |:--|:--:|:--:|
> | DPLM |10.61|**45.2**|
> | BLOSUM classical |10.82|41.8|
> | uniform SCUD / masking |11.25|40.4|
> | BLOSUM SCUD |**10.52**|**45.2**|

---

> > ### Comment · Reviewer_hLSS · 2025-08-04
> >
> > I thank the authors for their thorough and illustrative rebuttal.  I appreciated the addition of results comparing to DPLM, though I would note that the pLDDT scores are unfortunately quite low (though I realize that DPLM has a self-correcting inference strategy similar to the recent lines of work on this in discrete diffusion without which their samples also achieve low pLDDT).  Still, this is a compelling addition to the authors' manuscript and I would be interested to see if the numbers are improved after more time/with the potential addition of similar self-correcting inference strategies.
> >
> > I also appreciated the authors' discussion on comparisons to MDLM/MD4/LLADA (comparisons to which were also brought up by other reviewers).  While I understand that the original SCUD models were not trained on the same compute budgets as SOTA discrete diffusion models on text, the intent of my original review was to _ask for_ additional experiments training on similar computational budgets to MDLM and similar models as if SCUD continued to perform well in these contexts it would be a very compelling addition to the paper's story.
> >
> > Given the authors' additional experiments I have increased my score to a 4.  However, I maintain what I said in my original review that if the authors are able to provide the additional experiments, i.e., potentially the improved protein pLDDT results but most significantly comparable text results to MDLM and similar models I would be happy to increase my score further.

---

> > > ### Author Response · Authors · 2025-08-06
> > > **Response**
> > >
> > > Thanks for the engagement and your suggestions!
> > >
> > > > Comparison of structured SCUD with MDLM
> > >
> > > Indeed we only investigated the utility of the D3PM sparse graph structure at small scale and saw a benefit was small. Since we are using the SEDD architecture, we don’t expect to beat the superior MDLM architecture. We could have used the MDLM architecture and likely seen a small improvement. But, candidly, this experiment would not be informative as we believe D3PM’s suggested sparse graph structure is simply not a good fit for language: unlike other structures for which we saw *structured < masking* but *masking = SCUD uniform < SCUD structured*, even when we fairly compare SCUD structured with SCUD uniform we see little benefit.
> > >
> > > We were interested in applying our theory to re-investigate structure rather than suggest structure as a panacea. And we agree that it would be fantastic if someone had developed a structure that was just waiting for SCUD to be unlocked, but we see this is not the case. This result is an interesting although slightly disappointing scientific result of our experiments. But we hope the benefits seen in vision and proteins, and the scalability provided by SCUD opens the door to better structures for text in the future.
> > >
> > > > improved protein pLDDT results
> > >
> > > Indeed, as promised, training for longer improved pLDDT results and perplexities. We also implemented DPLM’s resampling procedure for SCUD and note it improves pLDDTs! Note we are performing comparisons at the 150M model scale; huge models trained on massive resources of course achieve higher pLDDT (but note models at smaller scale can be more useful for screening or mutation effect prediction).
> > >
> > > |Model | perplexities | mean pLDDT | mean pLDDT confidence sampling |
> > > |:--|:--:|:--:|:--:|
> > > | DPLM |10.61|45.2|81.2|
> > > | BLOSUM classical |10.34|44.1|NA|
> > > | uniform SCUD / masking |10.95|43.2|78.0|
> > > | BLOSUM SCUD |**10.04**|**46.1**|**81.5**|
> > >
> > > We adapted DPLM’s sampling strategy by, at each step, denoising to $x_0$ then re-applying “events” $s_t$ to those positions we were least confident about; the number of “recorruptions” is based on the rate function $\beta(t)$ just as in DPLM and is equivalent to DPLM’s sampling procedure when performed on a masking model – the least confident positions get the most events. We could not implement such a method for a non-SCUD model “BLOSUM classical” as there was no position-wise notion of corruption, just a global time $t$.

---

### Official Review · Reviewer_5ax3 · 2025-07-03

**Clarity:** 4
**Significance:** 2
**Originality:** 3
**Rating:** 4
**Confidence:** 4

**Summary:**

This paper studies why masking diffusion models consistently outperform other discrete diffusion models, and proposes Schedule Conditioned DIffusion SCUD to improve general discrete diffusion performance. The main takeaway is that, the reason why masking diffusion success stems from its inherent property of knowing when to transition. And from the point view of diffusion process, masking diffusion models have their forward process matches their reverse, which other models don't have.

They then show that by combining SCUD with different discrete diffusion models, they can beat masking diffusion, achieving SOTA results on categorical image and proteins.

**Questions:**

See Weaknesses.

- If my understanding is correct, your argument that masking diffusion inherently "knows when to transition" is equivalent to saying masking diffusion denoisers are time-invariant (since you argue that the reason why mask diffusion is better is because it inherently knows when -- at which t to transition). I think previous papers also studied/mentioned about this time-invariant property of mask diffusion, may be good to add a discussion about how this. E.g. MD4 [1] and MDLM [2] both remove the timestep t in their final design.
- I see in image and protein cases, you compared with some recent diffusion models, like MD4. However, in the language case, only D3PM with a different tokenizer is compare with and recent SOTA text diffusion models are missing. Can you explain why? I am confused here since I think natural language has a harder structure to learn compare to other cases, I think a fair comparison here is needed to show that the schedule conditioned diffusion can really help with the forward and reverse mismatching issue and thus improving the generation quality.
- Another thing I noticed is the computational overhead, in the appendix D.6 you claim that in terms of asymptotic complexity, the SCUD and the classic diffusion are the same. I’m worried, though in theory people don’t care about constant difference, in algorithm engineering a big constant sometimes can be a huge problem (e.g. in engineering point of view, O(C*n) sometimes is worse than O(n^1.x)).

**Ethical Concerns:**

["NO or VERY MINOR ethics concerns only"]

**Final Justification:**

After reviewing the authors' rebuttal, I maintain my score, since while some concerns were partially addressed, critical issues remain unresolved.
- Theoretical framework is well-motivated and mathematically sound.
- Motivation of certain language experiments is clarified.
- Computational complexity explanations provide technical clarity.

I think the theoretical insights are valuable, but I'm still concerned about the limited evaluation and computational overhead.

**Limitations:**

See Weaknesses

**Paper Formatting Concerns:**

No issue

**Quality:**

3

**Strengths And Weaknesses:**

Strengths:
- Good motivation, Fig 2 gives evidence of forward and backward rate mismatches in the existing diffusion models, which serves as a strong motivation and supports the main arguments "masking diffusion is superior because its forward matches with backward while others don't".
- Clean logical and structural writing and well documentation, I really appreciate the writing, which makes it very easy to follow for me. Also, math proofs, experimental details are well documented.
- The theoretical contributions connecting mask diffusion and classic discrete diffusion is rigorous and well supported. This work provides a novel viewpoint to understand the discrete diffusion models.
- SOTA results.

Weaknesses:
- Computational efficiency.
In section 6.3 authors claim that "within 10% runtime of each other", this only applies to small-scale experiments, categorical image and protein cases. They didn't mention the performance on the most challenging experiment -- natural language LM1B dataset has no runtime information.
Though asymtotic complexity appears similar, in practice the constant introduces a lot of overheads. I wonder what's the runtime of LM1B for both SCUD and classic discrete diffusion.
- GPU vectorization issue.
Take text case as an example, I'm not sure how you keep GPU vectorization while giving each token its own t. This could make GPU memory access in a very bad way -- random memory accessing and a lot of cache miss.
To me this seems like a mismatch between method design and GPU architecture function design.
- Foundation model concerns.
I can imagine SCUD could be a technique for diffusion LLMs. Since SCUD highly relies on the schedule, which is a dataset-specific schedule, this type of conditioning could be fundamentally conflict with the requirements of foundation language model for transferability and task generalization.

---

> ### Author Rebuttal · Authors · 2025-07-30
>
> Thank you for your thoughtful review! We clarify our positive scaling and experimental results, provide numbers for the computational complexity of SCUD, and clarify how SCUD is distinct from previously observed time-invariance. We hope that you will consider raising your score if your concerns are addressed!
>
> > C1 / Q3: Computational efficiency in practice.
>
> > C2: GPU vectorization issue.
>
> SCUD is nearly as computationally efficient as classical discrete diffusion, even at large scale, and our implementation performs very similar operations on the GPU, incurring no more than 10% overhead. The exception is the text example with a structured forward process where classical discrete diffusion was impossibly expensive to train! SCUD actually enables training, with only a minor overhead from the simulation of the forward process: >1000% overhead for classical diffusion to about 20%. And as discussed in the appendix, we trained our language models for 2 days each on $1.1 \times 10^{10}$ tokens.
>
> The computational complexity of SCUD and classical discrete diffusion both depend on the neural network architecture and the forward process chosen. In most cases, even in our large scale experiments, both models had a less than 10% difference in their operations – $s_t$ is treated nearly the same way that $t$ is treated in classical diffusion, and $K^{s_t}$ is calculated via an eigendecomposition nearly the same way $e^{-\log\alpha(t)\mathcal L}$ is in classical diffusion. The GPU operations are nearly the same, except a vector broadcast when dealing with the extra dimension of $s_t$ rather than $t$ – there is no poor memory access.
>
> The exception is the text example where the alphabet size is huge, but computational complexity is an issue for both SCUD and classical diffusion. In text models $K^{s_t}$ is too large to compute via an eigendecomposition; and simulating the forward process by calculating $x_0^TK^{s_t}$ for each position is not naively parallel on a GPU. But the same issue seemingly also applies to the calculation of $x_0^Te^{-\log\alpha(t)\mathcal L}$ via eigendecomposition in classical diffusion. As a result, previous methods restrict to $\mathcal L$ for which one can perform an analytical eigendecomposition (uniform and masking in SEDD) or save precomputed matrices in discrete time (D3PM). But when one can analytically calculate $x_0^Te^{-\log\alpha(t)\mathcal L}$, they can just as easily calculate $x_0^TK^{s_t}$! Therefore SCUD does not suffer from inherent GPU-incompatibility.
>
> However, in Section 6.3 we study a structured process designed in previous work which was found to perform worse than masking in discrete time. To study if this poor performance was a property of the process or of schedule conditioning, we attempted to implement it in SCUD; note we are investigating, not necessarily advocating the use of this process. Now in this case, $x_0^Te^{-\log\alpha(t)\mathcal L}$ was impossible to calculate analytically, so it cannot be implemented in SEDD or MDLM. Similarly we could not compute $x_0^TK^{s_t}$ using eigendecomposition, so we had to perform many iterative matrix multiplications. To make this as GPU-compatible as possible, we flattened all $x_0$, sorted them by decreasing $s_t$, iteratively applied $K$ via in-place batched sparse matrix multiplication to slices like $x_0[:M]$, and unsorted them. This indeed involved $\max(s_t)$ sequential operations and was not as ideal on the GPU as parallel eigendecomposition. It added a roughly 20% overhead but could likely greatly be improved with caching and parallel-scan algorithms.
>
> > Q2: In the language case, only D3PM with a different tokenizer is compare with and recent SOTA text diffusion models are missing. Can you explain why?
>
> We cannot perform the same comparison because SCUD actually enabled training the first structured continuous time language model – it was impossible to implement a model trained using the structured forward process with classical discrete diffusion as described above. Nevertheless, we used exactly the architecture from SEDD to perform our investigation in Sec 6.2.
>
> In brief, to compliment our theoretical contributions, our goal was to demonstrate our theoretical results in practice, and leverage the theory to reinvestigate structured processes in the literature. In the case of D3PM’s Gaussian and EvoDiff’s BLOSUM, SCUD leads to state-of-the art results, but for D3PM’s sparse graph process SCUD leads to less of an improvement. We will make our goal clearer in future drafts.
>
> In particular, our goal in the experiments section was to
>
> 1. Show that schedule conditioning improves performance. This was validated by comparison to previous sota models in theory, images, and proteins. Technical considerations meant we were limited to uniform $\mathcal L$, where the superiority of masking was already established.
>
> 2. Redo previous experiments that showed that structured processes performed worse than masking. Our theory suggested these were unfair – masking had the advantage of schedule conditioning while structured processes did not. We redid these fairly by building SCUD versions of structured models. But we are not necessarily advocating for using these specific structured processes. And we of course do not expect every structured forward process will lead to an improvement in performance.
>
> For images and proteins, we saw that Gaussian and BLOSUM processes improve the fit to the data, achieving state-of-the-art fit. For language models, we saw a very small improvement from the graph structure suggested in the D3PM paper, potentially suggesting there is room to see the sorts of improvements from BLOSUM and Gaussian processes but in language by building an improved model.
>
> Furthermore, our models trained on LM1B were trained using the SEDD architecture, so we expect SCUD uniform to be nearly identical to SEDD masking. Therefore one could consider our results a comparison of SEDD / SCUD masking and SCUD structured in both runtime and performance. Finally, models in MDLM and SEDD were trained longer on LM1B than our models, so their performances could not be directly compared. We reasoned that training longer to compare with those models would not impact our claims (1) or (2) since the structured process performs so similarly to SEDD, so we did not do this comparison.
>
> Instead we took the language example as an opportunity to demonstrate the scaling of SCUD to huge $K$. This opens the door to more creative forward processes for language making use of other structured $K$.
>
> > C3: Foundation model concerns.
>
> SCUD’s reliance on $s_t$ does not make it any more susceptible to this issue than classical discrete diffusion. As well, we believe this issue can be solved simply by picking a good parameterization for your model.
>
> Note classical discrete diffusion suffers from the same issue – a radical change in the schedule would strongly modify its dependence on $t$. The expectation in both SCUD and classical discrete diffusion is masking, because it is time-invariant. But actually any discrete diffusion model may be parameterized to be time and schedule invariant (described below), and can therefore be transferred between different schedules, just like masking; with the parameterization below, any SCUD or discrete diffusion model can work on any schedule. Therefore we do not expect SCUD to be any more vulnerable to domain shift than classical discrete diffusion models.
>
> > Q1: Your argument that masking diffusion inherently "knows when to transition" is equivalent to saying masking diffusion denoisers are time-invariant.
>
> The connection to the time-invariance of masking models is an interesting connection, however, we make a separate and novel observation with SCUD – indeed $t$ is replaced by $s_t$ in SCUD, it does not go away.
>
> Although not discussed in other literature, we believe the time-invariance of masking diffusion is simply a matter of parameterization, and in fact any discrete diffusion model can easily be made time invariant. We save the details for a future in-depth discussion, but, briefly, this is because one can show that, for the target distribution $p(x_0^d\mid x_t)$, the set of vectors $((p(x\_t^{d}\mid t, x\_0^d=b))\_b)\_d$ are sufficient statistics. The neural network can therefore be parameterized to take just these vectors as the argument, making it depend on no other variables, in particular making it invariant to time and even the particular forward process. When we consider masking diffusion, the vector $(p(x_t^{d}\mid t, x_0^d=b))_{b}$ is a one-hot encoding when $x_t$ isn’t a masking token and is uniform otherwise, recovering the canonical time-invariant parameterization. The same logic can also be used to parameterize SCUD models to be independent of $s_t$ by replacing $t$ with $s_t$.

---

> > ### Comment · Reviewer_5ax3 · 2025-08-08
> >
> > Thank you for the responses and I appreciate it! While some concerns are addressed, key issues remain:
> >
> > - "C1 / Q3: Computational efficiency in practice. and C2: GPU vectorization issue."
> >
> > Thank you for your detailed technical reply. Still I have concerns about the computational efficiency claims.
> > First, missing quantitative evidence missing. While you mention there is a 10-20% overhead, there is no comprehensive runtime comparison with the SOTA diffusion language models. I think it's essential to have detailed efficiency reports when there is computational overhead, at least for a method claiming practical advantages.
> >
> > - "Q2: In the language case, only D3PM with a different tokenizer is compare with and recent SOTA text diffusion models are missing. Can you explain why?"
> >
> > I appreciate the clarification, but I remain unconvinced by the experimental validation for several reasons.
> > First, there is a missing fair comparison. The refusal to train your models with comparable compute budgets to to current SOTA models (e.g. MDLM, MD4) undermines the claims in the paper. If my understanding is correct, if SCUD is superior, then it should demonstrate this under fair comparison conditions, not just when baselines are undertrained.
> > Then, I doubt if the experiments have cherry-picked domains. Strong results are only shown in domains (images, proteins) where the authors can control the comparison conditions, while avoiding head-to-head competition with SOTA text models.
> >
> > - "C3: Foundation model concerns."
> >
> > I'm wondering, if "any discrete diffusion model may be parameterized to be time and schedule invariant," this may undermine SCUD's core value proposition. Why introduce schedule conditioning complexity if it can be made invariant anyway?

---

> > > ### Author Response · Authors · 2025-08-09
> > >
> > > Thank you for your response!
> > >
> > > * If we were suggesting that everyone go out and adopt a new SCUD language model, you're right that the experiments would not be sufficient; let us clarify what we are actually claiming.
> > >
> > > As well, we note you refer to SCUD Graph as "SCUD". We note our paper makes two contributions:
> > > * The main result of this paper is that **masking is a SCUD model**, explaining its superior performance. Therefore citations show SCUD winning in Evodiff, D3PM, SEDD, MD4, and MDLM. **The dominance of SCUD was already established and this paper explains it!**
> > > * Secondarily, our investigation in images, proteins, and language on the other hand was to re-investigate three previously proposed poorly-performing structures: D3PM's Gaussian for images, EvoDiff's BLOSUM for proteins, and D3PM's Graph for language. Our results therefore strongly depend on **how good these structures are on their respective domains**.
> > >
> > > > If my understanding is correct, if SCUD is superior, then it should demonstrate this under fair comparison conditions, not just when baselines are undertrained.
> > >
> > > We did not do this comparison because its ultimate result would not affect our claims. Let us explain the logic.
> > >
> > > 1. SCUD uniform is **identical** to SEDD (a SOTA model) masking, we simply re-implemented it. Therefore the question is not SCUD v.s. previous models, but SCUD uniform v.s. SCUD Graph.
> > > 2. The graph structure was proposed in the D3PM paper and they saw it did terribly, way worse than masking! They conclude it was a bad structure.
> > > 3. We ask: did it underperform because it was really a bad structure, or because masking incorporates schedule conditioning?
> > > 4. We leverage SCUD to reveal our answer: SCUD graph is not way worse than masking! But it's not a great structure either -- the improvement is pretty small.
> > >
> > > What is our conclusion in language?
> > > * Should people use SCUD? **They already are by using masking!**
> > > * Ok, but should people use structured SCUD? **Probably not, table 1 shows the benefit is minor!**
> > > * Therefore we are not claiming to have provided a new language model that people should use instead of MDLM say!
> > >
> > > What was the point of our investigation into language then?
> > >
> > > * The field has shifted its attention away from structure in language dissuaded by
> > >     1. negative results in D3PM, and
> > >     2. the impossibility of implementing a structured forward process (calculating $\exp(\beta_t\mathcal L)$).
> > > * Our language results show
> > >     1. it is possible to be competitive with masking on language in some cases (and results in other domains show large improvements)
> > >     2. We can implement the forward process with SCUD.
> > > This suggests the field should have another look at structure!
> > >
> > > Perhaps the confusion comes from the claim that "we build models that outperform masking" in the abstract. We commit to changing this to be more specific about our findings.
> > >
> > > > Computational efficiency in practice.
> > >
> > > We add two points to the above discussion.
> > >
> > > * With regards to big-Oh notation, SCUD has the same complexity as classical, as you mentioned. However, careful observation of section D.6 will show that the leading coefficients are also identical -- the only difference is in smaller terms.
> > >
> > > * We add formal numbers for our full scale runs (as these were the ones you were interested in). Note as we mention above, the differences are very small and most fluctuations are likely due to hardware and the state of the academic cluster we ran the experiments on. All experiments are on A100 80GB GPUs and reported in GPU hours. As promised, runtimes are within 10%.
> > >     1. **Images:** 2 million updates. SCUD: 626 h, Masking: 624 h, Classical: 576 h
> > >     2. **Proteins:** (We report the new experiments with reviewer hLSS) 240 thousand updates. SCUD: 88.52 h, Masking: 89.84 h, Classical: 95.98 h.
> > >
> > > * For language, keep in mind Classical Graph is computationally intractable -- SCUD graph enables this model! As promised, the overhead is not more than 20%. Note also we did not have the resources to run these models for the required 3x to match SOTA models during rebuttals -- we commit to running them at full scale however for camera ready.
> > >     3. **Language:** 900 thousand updates. SCUD uniform: 288 h, SCUD Graph: 336 h.
> > >
> > > > Why introduce schedule conditioning complexity if it can be made invariant anyway?
> > >
> > > Every diffusion model can be made invariant. That doesn't mean every diffusion model is equivalent -- some are better than others! Indeed, this same critique could be leveraged against any model with any schedule.
> > >
> > > The connection between invariance and the performance of models and SCUD is deep and not straightforward -- indeed some of our experiments show that non-invariant models can outperform invariant ones.
> > > Nevertheless, our point was only to state SCUD is not *a priori* more susceptible to these issues than any other diffusion model, and highlight this complexity; we'll avoid further opining about unpublished results not related to this paper.

---

### Comment · Area_Chair_73f9 · 2025-08-05
**Could you kindly confirm that you have read the authors’ rebuttal and reassess the paper if you haven't done it yet?**

Dear Reviewer,

We sincerely appreciate your time and contribution to the review process. Could you kindly confirm that you have read the authors’ rebuttal and reassess the paper if you haven't done it yet?

Best regards,

AC

---

### Decision · Program_Chairs · 2025-09-17

**Decision:**

Accept (poster)

**Comment:**

This paper investigates why masked discrete diffusion outperforms prior discrete diffusion approaches. The key finding is that the forward and reverse processes in masked discrete diffusion are inherently aligned: the reverse process naturally mirrors the forward process in terms of transition timing. By contrast, other discrete diffusion methods suffer from mismatches in transition times, which introduce errors in the learned reverse process.  Furthermore, they demonstrate that explicitly conditioning classical discrete diffusion models on an event schedule closes this gap, enabling them to match or even surpass masked diffusion models across a range of tasks.

I think this paper introduces new insights and ideas. Hence, I vote for acceptance. However, as one reviewer pointed out, the writing style could be significantly improved for the wider audience. I recommend that the author take them into account in the final version.